# A multi-phenotype analysis reveals 19 susceptibility loci for basal cell carcinoma and 15 for squamous cell carcinoma

Mathias Seviiri [1,2,3] ✉, Matthew H. Law [1,2], Jue-Sheng Ong [1], Puya Gharahkhani [1], Pierre Fontanillas [4], The 23andMe Research Team*, Catherine M. Olsen [5,6], David C. Whiteman [6] & Stuart MacGregor [1,2]

Basal cell carcinoma and squamous cell carcinoma are the most common skin cancers, and have genetic overlap with melanoma, pigmentation traits, auto-immune diseases, and blood biochemistry biomarkers. In this multi-trait genetic analysis of over 300,000 participants from Europe, Australia and the United States, we reveal 78 risk loci for basal cell carcinoma (19 previously unknown and replicated) and 69 for squamous cell carcinoma (15 previously unknown and replicated). The previously unknown risk loci are implicated in cancer development and progression (e.g. *CDKL1*), pigmentation (e.g. *TPCN2*), cardiometabolic (e.g. *FADS2*), and immune-regulatory pathways for innate immunity (e.g. *IFIH1*), and HIV-1 viral load modulation (e.g. *CCR5*). We also report an optimised polygenic risk score for effective risk stratification for keratinocyte cancer in the Canadian Longitudinal Study of Aging (794 cases and 18139 controls), which could facilitate skin cancer surveillance e.g. in high risk subpopulations such as transplantees.

Keratinocyte cancers (KC), including basal cell carcinoma (BCC) and squamous cell carcinoma (SCC), are the most commonly diagnosed cancers globally. KC resulted in over 5.4 million diagnoses and $8 billion dollars in expenditure in the US in 2011 alone[1], while in Australia, they account for >24% of all-cancer diagnoses[2], and impose a huge economic burden on the health sector costing over AUD $700 million for treatment annually[3]. KC is responsible for up to 8700 deaths a year in the United States[4]. The relative rates and morbidity, from KC, is even higher in Australia[5]. BCC and SCC share many common risk factors, including sun exposure, skin and hair pigmentation and immunosuppression.

Skin cancers, and pigmentation traits and autoimmune diseases have several susceptibility genes overlapping[6–9]. For example, several variants in pigmentation genes *ASIP/RALY, IRF4, MC1R, OCA2, SLC45A2* and *TYR*, are associated with BCC, SCC and melanoma[8,10]. Shared

immune-regulatory genes in the *HLA* and *LPP* regions have been found to influence susceptibility to BCC, SCC, melanoma and autoimmune diseases such as rheumatoid arthritis, vitiligo, type 1 diabetes and psoriasis[6–9]. There are also some tumour-genesis-related genes, which are expressed in both KC and other non-skin cancers. For example, oncogene *TNS3*, which is overregulated in BCC, is also associated with breast, lung and prostate cancers[8,11,12]. Furthermore, *HAL* at 12q23.1 has been found to be associated with KC risk[13] as well as vitamin D levels[14]. However, standard single GWAS meta-analysis approaches are unable to utilise this multi-trait genetic overlap to further explore the genetic risk for BCC, and SCC.

Multivariate GWAS approaches, such as multi-trait analysis of GWAS (MTAG)[15], can draw on this overlapping genetics to identify new risk regions (here for BCC or SCC). MTAG is a generalisation of inverse-variance-weighted meta-analysis that importantly accounts for

[1]Statistical Genetics Lab, QIMR Berghofer Medical Research Institute, Brisbane, QLD, Australia. [2]School of Biomedical Sciences, Faculty of Health, Queensland University of Technology, Brisbane, QLD, Australia. [3]Center for Genomics and Personalised Health, Queensland University of Technology, Brisbane, QLD, Australia. [4]23andMe, Inc, Sunnyvale, CA, USA. [5]Cancer Control Group, QIMR Berghofer Medical Research Institute, Brisbane, QLD, Australia. [6]Faculty of Medicine, University of Queensland, Brisbane, QLD, Australia. *A list of authors and their affiliations appears at the end of the paper.
✉ e-mail: Mathias.Seviiri@qimrberghofer.edu.au

incomplete genetic correlation, and sample overlap, between GWAS. A key property of MTAG is that it outputs estimates of trait-specific effect sizes and p-values for each of the input traits—in this case BCC or SCC. We have previously used MTAG to identify loci for KC based on the genetic correlation between BCC and SCC only[13]. BCC and SCC are different in terms of polygenicity and aetiology and therefore, we sought to identify susceptibility genetic loci for BCC and SCC by exploring their genetic overlap with melanoma, pigmentation traits, autoimmune diseases, and blood biochemistry biomarkers in a multi-phenotype analysis of GWAS.

In this work, we show that BCC and SCC have a high genetic correlation with melanoma, pigmentation traits, autoimmune diseases, and blood biochemistry biomarkers. We use MTAG to leverage this genetic overlap and identify 78 and 69 independent genome-wide significant susceptibility loci for BCC and SCC, respectively; 19 BCC and 15 SCC loci are both previously unknown and replicated in a large independent cohort. The previously unknown risk loci are implicated in BCC/SCC development and progression, pigmentation, cardiometabolic pathways, and immune-regulatory pathways, including; innate immunity, HIV-1 viral load modulation and disease progression. We also report a optimised BCC polygenic risk score (PRS) that enables effective risk stratification for KC.

## Results
### Genetic correlation
Using linkage disequilibrium score (LDSC) regression[16], 20 phenotypes were significantly genetically correlated ($P < 0.05$, $rg > 10\%$) with either BCC or SCC (Fig. 1 and Supplementary Data 1). In the first instance, 35

phenotypes that we considered as possibly correlated with skin cancer (including body mass index) were excluded for not meeting the aforementioned criteria above (Supplementary Data 2). Using the same selection criteria, no additional new phenotypes were included following analysis using collated GWAS summary statistics (over 700 phenotypes) in the LD hub database[17]. In total, subsequent analyses included 22 genetically correlated traits; cancers; BCC and SCC GWAS from the UK Biobank (UKB)[18,19], a cutaneous melanoma GWAS meta-analysis[20], KC from the QSkin Sun and Health Study (QSkin)[21], KC from the Electronic Medical Records and Genomics Network (eMERGE) cohort[22,23] and all-cancer from the Resource for Genetic Epidemiology Research on Aging (GERA) cohort;[24] skin and hair pigmentation related traits; skin burn type (QSkin), red hair (QSkin), hair colour excluding red hair (UKB), skin colour (UKB), and mole count excluding mela-noma cases (QSkin), autoimmune conditions; type 1 diabetes and hypothyroidism[25], and vitiligo[26], lifestyle-related traits; educational attainment in years spent in school[27] and smoking (cigarettes per day)[28], and biochemistry blood biomarkers from the UKB; aspar-tate aminotransferase, C-reactive protein, albumin, and gamma-glutamyl transferase, glucose and vitamin D (adjusted for monthly variation). The sample sizes and phenotype measurements for all the included and excluded traits are presented in Supplementary Data 3 and 2, respectively.

### Discovery of genome-wide significant susceptibility loci for BCC and SCC
Adding 20 traits genetically correlated with either BCC or SCC ($r_g > 0.1$, $P < 0.05$) (from UKB) increased the effective sample sizes for BCC and

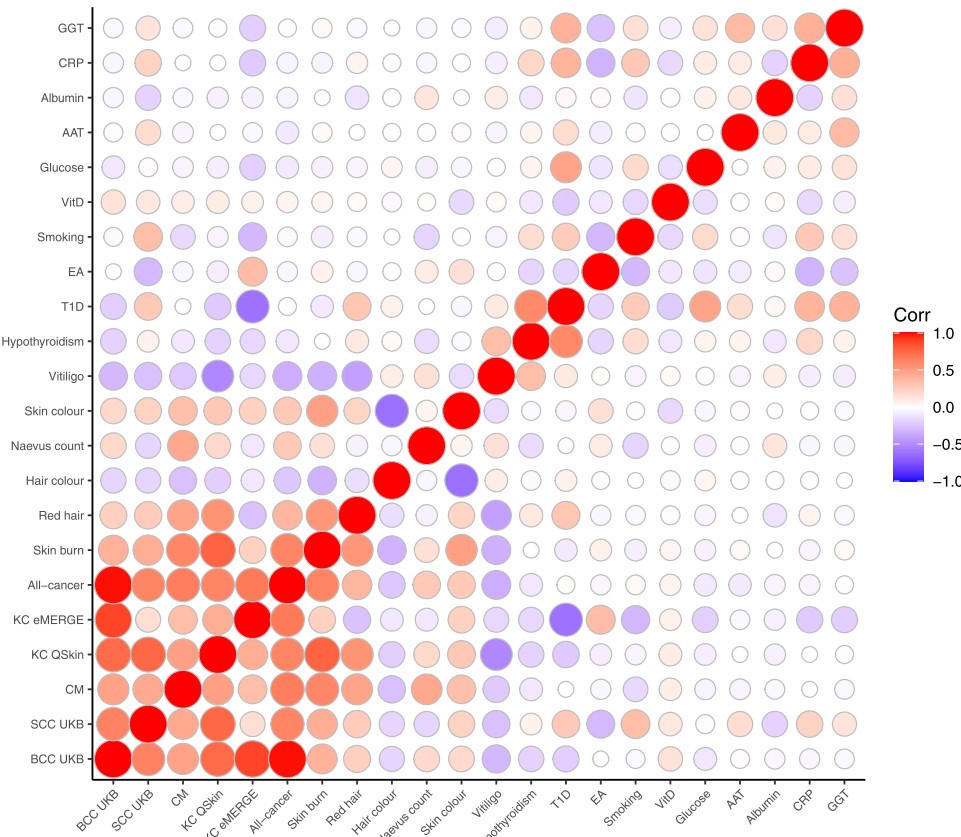

**Fig. 1 | Heatmap for the genetic correlation between 22 traits with a significant correlation with either BCC or SCC.** Bivariate genetic correlation 22 traits that were significantly correlated ($P < 0.05$, rg >10%) with the UKB BCC or SCC GWAS. BCC UKB basal cell carcinoma in the UK Biobank, SCC UKB squamous cell carci-noma in the UK Biobank, CM cutaneous melanoma, KC QSkin keratinocyte cancer in the QSkin cohort, KC eMERGE keratinocyte cancer in the eMERGE cohort, Hypothyr hypothyroidism, T1D type 1 diabetes, EA education attainment, VitD vitamin D, AAT aspartate aminotransferase, CRP C-reactive protein, GGT gamma-glutamyl transferase and Corr correlation. Source data are provided as a Source Data file.

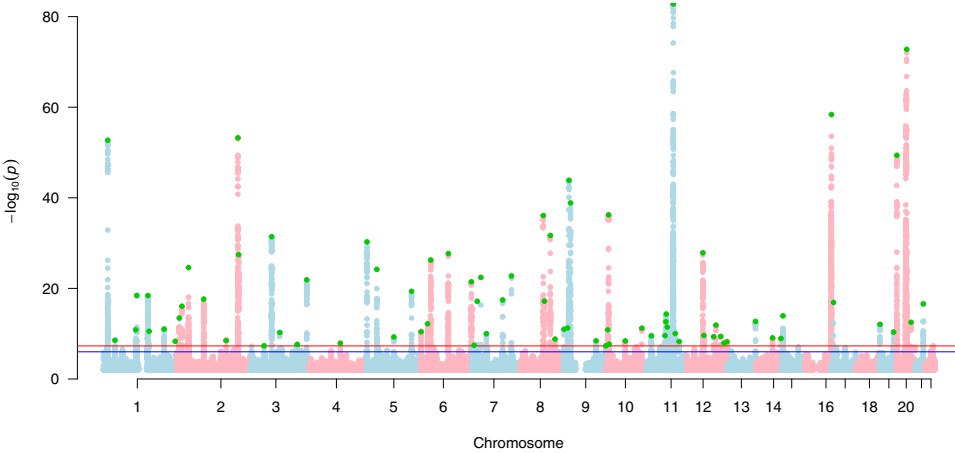

**Fig. 2 | Manhattan plot for basal cell carcinoma susceptibility.** The Manhattan plot shows the association between SNPs and basal cell carcinoma susceptibility based on the MTAG approach. The Y-axis represents the level of significance recorded in negative log 10 (*P* value) (two-tailed test), whilst the X-axis represents the chromosome 1–22, alternated with light blue and light pink colours. The horizontal blue line represents a suggestive level of significance at *P* value = $10^{-6}$, while the red one represents the genome-wide level of significance; $P = 5 \times 10^{-8}$. The green dots represent the 78 genome-wide significant independent loci for basal cell carcinoma susceptibility (after multiple correction for a million tests; 0.05/1,000,000). Only SNPs with a *P* value <0.01 were included. The source data file is provided as the BCC summary statistics in the GWAS Catalogue under accession code GCST90137411.

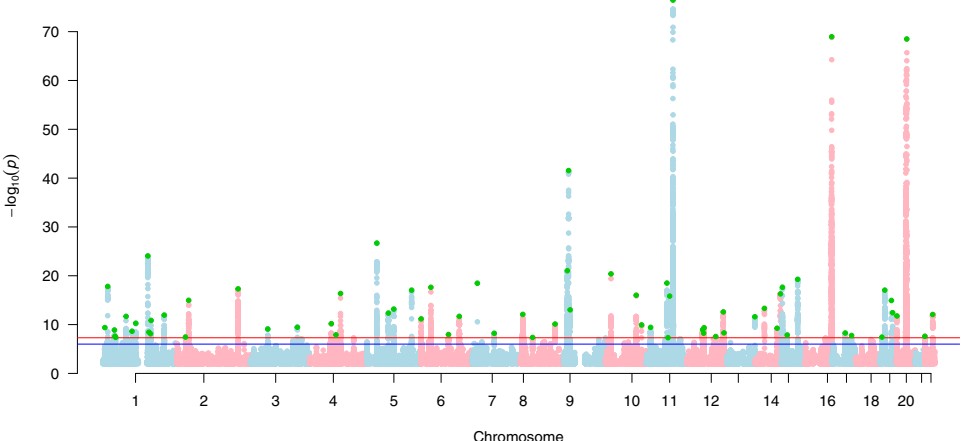

**Fig. 3 | Manhattan plot for squamous cell carcinoma susceptibility.** The Manhattan plot shows the association between SNPs and squamous cell carcinoma susceptibility based on the MTAG approach. The Y-axis represents the level of significance recorded in negative log 10 (*P* value) (two-sided test), whilst the X-axis represents the chromosome 1–22, alternated with light blue and light pink colours. The horizontal blue line represents a suggestive level of significance at *P* value = $10^{-6}$, while the red one represents the genome-wide level of significance; $P = 5 \times 10^{-8}$. The green dots represent the 69 genome-wide significant independent loci for squamous cell carcinoma susceptibility (after multiple correction for a million tests; 0.05/1,000,000). Only SNPs with a *P* value <0.01 were included. The source data file is provided as the BCC summary statistics in the GWAS Catalogue under accession code GCST90137412.

SCC by 2.6 and 8.3 times, respectively. Using the MTAG approach, we identified 78 and 69 independent genome-wide significant ($P < 5 \times 10^{-8}$) susceptibility loci for BCC (Fig. 2 and Supplementary Data 4) and SCC (Fig. 3 and Supplementary Data 5), respectively. Although the results for the peak single nucleotide polymorphisms (SNPs) were more significant following the MTAG analysis due to the greater statistical power, the log (odds ratio) effect sizes for the MTAG output and the respective UKB BCC or SCC GWAS inputs were highly concordant. For BCC the Pearson's correlation of effect sizes was 0.93 (95% confidence interval [CI] = 0.89–0.96, $P < 2.20 \times 10^{-16}$; Fig. 4a). Similarly, concordance was high for SCC loci (Pearson's correlation = 0.71, 95% CI = 0.57–0.81, $P = 7.34 \times 10^{-12}$; Fig. 4c).

In the 23andMe, Inc replication sample (252,931 cases and 2,281,246 controls), 71 of the 78 susceptibility loci for BCC replicated at the genome-wide level ($P < 5 \times 10^{-8}$), 74 replicated after Bonferroni correction ($P = 6.49 \times 10^{-4}$), and 77 loci replicated at a nominal $P = 0.05$

(Supplementary Data 4). There was high concordance with the BCC effect estimates between the MTAG and the replication set with Pearson's correlation = 0.97 (95% CI = 0.95–0.98, $P = 2.20 \times 10^{-16}$; Fig. 4b). Of the 69 susceptibility loci for SCC, 25 replicated at the genome-wide level ($P = 5 \times 10^{-8}$), 31 replicated after Bonferroni correction ($P = 7.24 \times 10^{-4}$) and 38 loci replicated at a nominal $P = 0.05$ in the 23andMe cohort (135,214 cases and 2,404,735 controls) (Supplementary Data 5). For SCC, there was also high concordance with the effect estimates between the MTAG and the replication set with Pearson's correlation = 0.69 (95% CI = 0.55–0.80, $P = 3.48 \times 10^{-11}$; Fig. 4d).

**Description of the previously unknown loci for BCC and SCC**

A locus was considered previously unknown for BCC or SCC if it had not been significantly associated with either BCC, SCC or KC at the genome-wide level ($P < 5 \times 10^{-8}$) before, and if it replicated at minimum $P < 0.05$) in the 23andMe replication cohort. By this criterion, we

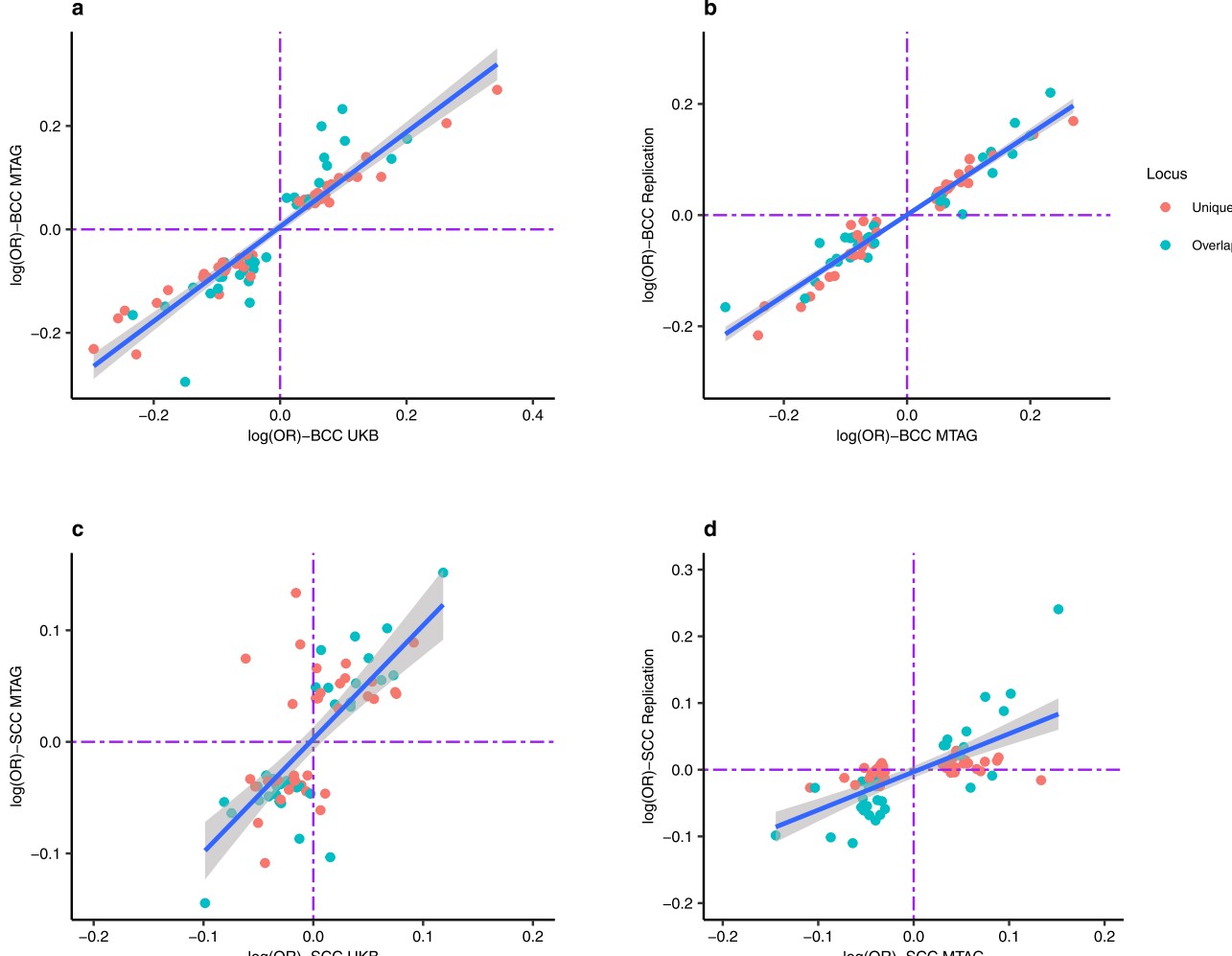

**Fig. 4 | Concordance of the log (OR) effect estimates for MTAG versus UK single-trait GWAS and 23andMe replication.** Figure 4 shows the comparison of the effect estimates in log (odds ratio) for both basal cell carcinoma (BCC) and squamous cell carcinoma (SCC) based on the respective MTAG approach results versus UKB single-trait GWAS and replication results from 23andMe. The blue line is the line of best fit with the 95% confidence intervals. The blue dots represent loci that overlap between BCC and SCC, whilst the red dots show the loci that are respectively unique to BCC or SCC. The dotted purple lines represent null effects (i.e. log (OR) = 0). The Y- and X- axes represent log (OR). **a** Shows BCC MTAG versus UKB BCC effect estimates, yielding a high concordance with a Pearson's correlation of 0.93 (95% confidence interval [CI] = 0.89–0.96, two-sided test). **b** Shows BCC MTAG versus BCC replication (23andMe) effect estimates, yielding a high concordance i.e. Pearson's correlation = 0.97 (95% CI = 0.95–0.98, two-sided test). **c** Shows SCC MTAG versus UKB SCC effect estimates, yielding a high concordance; Pearson's correlation = 0.71, 95% CI = 0.57–0.81, two-sided test). **d** Shows SCC MTAG versus SCC replication effect estimates, also resulting in a high correlation i.e. 0.69 (95% CI = 0.55–0.80, two-sided test). UKB- United Kingdom Biobank, and MTAG- multi-trait analysis of GWAS. Source data are provided as a Source Data file.

identified 19 and 15 previously unknown loci for BCC (Table 1) and SCC (Table 2), respectively. The previously unknown loci were annotated to the pigmentation, cardiometabolic, cancer development/progression and immune-regulatory pathways (Figs. 5, 6), whilst others are known loci for cutaneous melanoma susceptibility (*ATM*, and *SOX6* for BCC, and *GPR98*, and *DSTYK* for both BCC and SCC). More details on these loci and the broader biological groups have been discussed in the Supplementary Information (Supplementary Note 1). For loci that are unique to BCC or SCC, or overlap between BCC and SCC, refer to Tables 1, 2.

## Gene-set pathways

After multiple correction testing ($P = 0.05/18,188$ genes; $2.75 \times 10^{-6}$), gene-set analysis revealed curated and gene ontology (GO) pathways that are important in the development of keratinocyte cancer (Supplementary Table 1). A number of pathways are involved in melanogenesis (e.g. melanin biosynthesis, melanin biosynthetic process and melanosome membrane); a process which influences the nature of pigmentation traits and response to UV exposure. Genes in the

"response to trabectedin" pathway are likely to play an important role in DNA damage response. Trabectedin is an alkylating agent used to treat certain cancers resulting in DNA damage. Other pathways are important in the downregulation of the immune response (e.g. GO negative regulation of regulatory T cell differentiation), and enhancement of the immune response (IL2-PI3K pathway, MHC class II receptor activity, and nuclear factor of activated T cells (NFAT) pathway for development and function of regulatory T cells).

## BCC MTAG-derived polygenic risk score for KC prediction in the Canadian longitudinal study on aging (CLSA)

During the validation of the PRSs, S5 (i.e. $P < 10^{-4}$ with 273 SNPs for the $MTAG_{PRS}$ and 462 SNPs for the $UKB_{PRS}$) was the optimal PRS models for both $MTAG_{PRS}$ and $UKB_{PRS}$ with Nagelkerke $R^2$ of 10.65 and 9.55% respectively (Fig. 7a). The total number of SNPs in both PRS was different because the MTAG results have more power than the single BCC analysis and therefore it has more SNPs reaching significance. However, based on 'the nearest gene' analysis, 154 SNPs (Supplementary Data 8) overlapped between the $MTAG_{PRS}$ and $UKB_{PRS}$. The correlation

**Table 1 | BCC susceptibility novel loci that replicated at *P* < 0.05 in 23andMe cohort**

| rsID | Locus | CHR | BP | EA | NEA | EAF | log (OR) | SE | *P* value | Nearest gene(s) | eQTL gene (skin) | eQTL gene *P* value |
|------|-------|-----|-----|-----|-----|-----|----------|-----|-----------|-----------------|------------------|---------------------|
| rs12142181 | 2 | 1 | 41,912,985 | T | C | 0.08 | −0.092 | 0.015 | 2.86E-09 | *EDN2* (−31kb) \| *FOXO6* (+63 kb) | *EDN2*** | 6.26E-05 |
| rs2369633 | 7 | 1 | 205,181,062 | C | T | 0.10 | −0.100 | 0.015 | 9.81E-12 | *DSTYK* (+0.33 kb) | *CNTN2* | 4.96E-06 |
| rs2111485 | 13 | 2 | 163,110,536 | G | A | 0.40 | −0.052 | 0.009 | 3.46E-09 | *FAP* (+10 kb) \| *IFIH1* (−13 kb) | – | – |
| rs2373232 | 16 | 3 | 46,444,383 | A | G | 0.32 | −0.050 | 0.009 | 4.77E-08 | *CCRL2* (−4 kb) \| *CCR5* (+26 kb) | *CCR2/CCR5* | 8.61E-6/7.88E-9 |
| rs9878566 | 19 | 3 | 156,493,213 | T | C | 0.48 | 0.048 | 0.009 | 2.36E-08 | *LINC00886* (0) | *LINC00886* | 1.56E-24 |
| rs6889986 | 24 | 5 | 90,207,399 | A | G | 0.44 | −0.054 | 0.009 | 5.44E-10 | *GPR98* (0) | – | – |
| rs706779 | 47 | 10 | 6,098,824 | C | T | 0.48 | 0.058 | 0.009 | 1.43E-11 | *IL2RA* (0) | *FHIT* (eQTL Gen) | 8.16E-07 |
| rs7098111 | 51 | 10 | 119,573,178 | T | C | 0.15 | −0.080 | 0.012 | 6.19E-12 | *RAB11FIP2* (−191 kb) | – | – |
| rs10766301 | 52 | 11 | 16,217,413 | T | C | 0.38 | 0.054 | 0.009 | 2.93E-10 | *SOX6* (0) | – | 6.15E-11 |
| rs174570 | 53 | 11 | 61,597,212 | T | C | 0.16 | −0.081 | 0.013 | 2.67E-10 | *FADS2* (0) \| *FADS1* (+12 kb) | – | – |
| rs2924552 | 56 | 11 | 68,889,367 | T | C | 0.41 | 0.061 | 0.009 | 3.50E-12 | *TPCN2* (+31 kb) | *TPCN2* (eQTL Gen) | 2.18E-5/8.94E-157 |
| rs4409785 | 58 | 11 | 95,311,422 | C | T | 0.19 | −0.073 | 0.011 | 9.51E-11 | *FAM76B* (−190 kb) | *SESN3* (eQTL Gen) | 5.37E-62 |
| rs73008229 | 59 | 11 | 108,187,689 | A | G | 0.16 | −0.071 | 0.012 | 5.40E-09 | *ATM* (0) | *C11orf65* | 3.18E-05 |
| rs10876864 | 61 | 12 | 56,401,085 | A | G | 0.41 | 0.055 | 0.009 | 2.56E-10 | *SUOX* (+1.78 kb) \| *IKZF4* (−13 kb) | *RPS26/IKZF4/SUOX* | 3.92E-251/9.86e-9/3.97e-54 |
| rs17425489 | 62 | 12 | 89,015,138 | A | G | 0.09 | −0.091 | 0.015 | 4.54E-10 | *KITLG* (+40 kb) | – | |
| rs10774625 | 64 | 12 | 111,910,219 | G | A | 0.48 | 0.053 | 0.009 | 4.03E-10 | *ATXN2* (0) | *ALDH2 (blood)* | 1.58E-08 |
| rs11059675 | 65 | 12 | 122,668,326 | A | G | 0.44 | −0.049 | 0.009 | 1.21E-08 | *LRRC43* (0) \|*IL31*(+9 kb) | *LRRC43* | 6.92E-06 |
| rs7301141 | 66 | 12 | 133,138,503 | G | A | 0.46 | 0.052 | 0.009 | 6.21E-09 | *FBRSL1* (0) | *P2RX2* | 4.76E-06 |
| rs1136165 | 69 | 14 | 103,988,180 | T | G | 0.38 | 0.054 | 0.009 | 1.11E-09 | *CKB* (0) | *CKB/MARK3* | 1.88E-22/3.15E-31 |

Genes are formatted in italics.
*CHR* chromosome, *BP* base pair position, *EA* effect allele, *NEA* non-effect allele, *EAF*- effect allele frequency, *OR* odds ratio (two-sided test), *SE* standard error, *P* P value after multiple correction test for a million tests (0.05/1,000,000), *eQTL* expression quantitative trait loci.

**Table 2 | SCC susceptibility novel loci that replicated at *P* < 0.05 in 23andMe cohort**

| rsID | Locus | CHR | BP | EA | NEA | EAF | log (OR) | SE | *P* | Nearest gene | eQTL gene (skin) | eQTL gene *P* value |
|------|-------|-----|-----|-----|-----|-----|----------|-----|-----|--------------|------------------|---------------------|
| rs3768321 | 3 | 1 | 40,035,928 | T | G | 0.19 | 0.041 | 0.007 | 1.32E-09 | *PABPC4* (0) | *PABPC4* | 2.65E-28 |
| rs12142181 | 4 | 1 | 41,912,985 | T | C | 0.08 | −0.054 | 0.010 | 2.41E-08 | *EDN2* (−31.46 kb) | *EDN2* | 6.26E-05 |
| rs17391694 | 6 | 1 | 78,623,626 | T | C | 0.10 | 0.054 | 0.008 | 2.12E-12 | *GIPC2* (+20.51 kb) | *FUBP1* | 5.49E-15 |
| rs3851290 | 13 | 1 | 205,149,508 | T | C | 0.39 | −0.039 | 0.006 | 1.17E-12 | *DSTYK* (0) | *DSTYK* | 5.52E-18 |
| rs1260326 | 14 | 2 | 27,730,940 | C | T | 0.41 | −0.030 | 0.005 | 3.57E-08 | *GCKR* (0) | *NRBP1* | 3.09E-18 |
| rs9878566 | 18 | 3 | 156,493,213 | T | C | 0.48 | 0.034 | 0.005 | 3.52E-10 | *LINC00886* (0) | *LINC00886* | 1.56E-24 |
| rs6889986 | 24 | 5 | 90,207,399 | A | G | 0.44 | −0.041 | 0.005 | 6.75E-14 | *GPR98* (0) | – | 8.94E-11 |
| rs77758638 | 33 | 8 | 42,014,917 | T | C | 0.11 | 0.045 | 0.008 | 4.35E-08 | *AP3M2* (0) | *POLB/AP3M2* | 4.39E-26 |
| rs35563099 | 40 | 10 | 119,572,403 | T | C | 0.15 | −0.046 | 0.007 | 1.19E-10 | *RAB11FIP2* (−192kb) | – | |
| rs2924552 | 42 | 11 | 68,889,367 | T | C | 0.41 | 0.049 | 0.005 | 3.19E-19 | *TPCN2* (+31.3 kb) | *TPCN2* | 2.18E-05 |
| rs10899466 | 44 | 11 | 78,013,674 | A | G | 0.18 | −0.061 | 0.007 | 1.47E-16 | *GAB2* (0) | *GAB2* | 1.24E-13 |
| rs10876864 | 47 | 12 | 56,401,085 | A | G | 0.41 | 0.031 | 0.005 | 5.95E-09 | *SUOX* (+1.78 kb) \| *IKZF4* (−13 kb) | *SUOX/IKZF4* | 3.97E-54/9.86E-9 |
| rs142004400 | 53 | 14 | 50,829,560 | C | A | 0.03 | −0.109 | 0.014 | 4.83E-14 | *CDKL1* (0) | *CDKL1 (eQTL Gen)* | 1.13E-19 |
| rs10141120 | 55 | 14 | 103,923,008 | C | T | 0.35 | −0.047 | 0.006 | 5.18E-17 | *MARK3* (0) | *MARK3/CKB* | 7.60E-39/3.33E-31 |
| rs472385 | 57 | 15 | 44,186,844 | A | G | 0.25 | −0.035 | 0.006 | 1.37E-08 | *FRMD5* (0) | *AC011330.5* | 5.74E-19 |

Genes are formatted in italics.
*CHR* chromosome, *BP* base pair position, *EA* effect allele, *NEA* non-effect allele, *EAF* effect allele frequency, *OR* odds ratio (two-sided test), *SE* standard error, *P* P value after multiple correction test for a million tests (0.05/1,000,000), *eQTL* expression quantitative trait loci.

of the effect size for the PRS SNPs across the two sets was consistent or high (e.g. for the overlapping 154 SNPs, Pearson's correlation = 0.94, 95% CI = 0.92–0.96, $P < 2.2 \times 10^{-16}$), meaning the extra MTAG SNPs are consistent but just better powered.

The SNPs for the optimal models are presented in Supplementary Data 6 and Supplementary Data 7 for the UKB$_{PRS}$ and MTAG$_{PRS}$, respectively. When we tested the performance for both the UKB$_{PRS}$ and MTAG$_{PRS}$ in the CLSA ($N = 18,933$), the MTAG$_{PRS}$ outperformed the

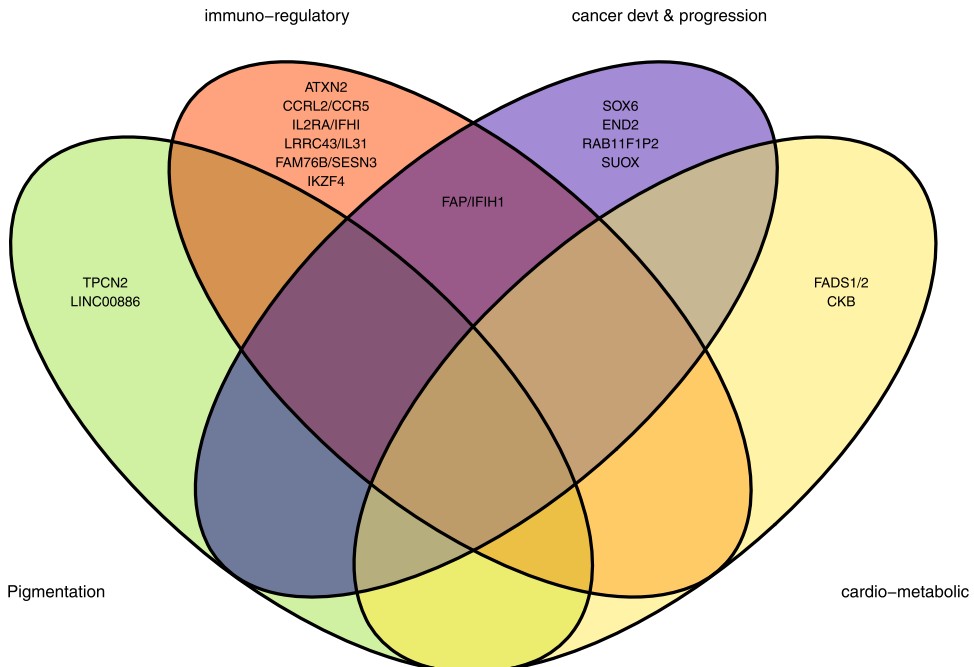

**Fig. 5 | Basal cell carcinoma loci and biological pathways.** The broad biological pathways included; pigmentation, immuno-regulatory, cardiometabolic and cancer development and progression. *FBRSL1*, *KITLG, ATM, GPR98* and *DSTYK* are not shown in this figure.

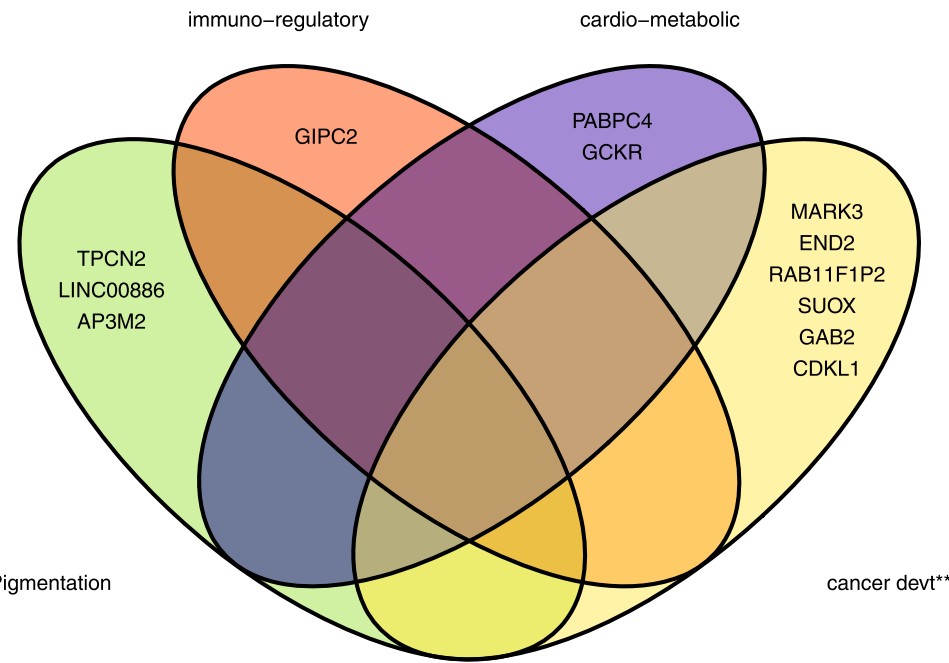

**Fig. 6 | Squamous cell carcinoma loci and biological pathways.** The broad biological pathways included; pigmentation, immuno-regulatory, cardiometabolic and cancer development and progression (cancer devt**). *GPR98* and *DSTYK* are not shown in this figure.

$UKB_{PRS}$ in terms of association with KC risk, KC risk prediction, and stratification. For example, after adjusting for age at recruitment, sex and the first ten PCs, the $MTAG_{PRS}$ outperformed the $UKB_{PRS}$ for association with KC risk i.e. $MTAG_{PRS}$ OR = 1.66, 95% CI = 1.55–1.79, $P = 1.95 \times 10^{-41}$ versus $UKB_{PRS}$ OR = 1.56, 95% CI = 1.45–1.67, $P = 3.38 \times 10^{-33}$ (Fig. 7b). In addition, the net reclassification index for KC risk was greater for $MTAG_{PRS}$ than the $UKB_{PRS}$ (Fig. 7c), when added to the base model containing age, sex and ten PCs. Consequently, the $MTAG_{PRS}$ compared to the $UKB_{PRS}$ reclassified more participants for KC risk to the appropriate risk group (low risk, moderate risk and high risk) (i.e. percentage of people reclassified; $MTAG_{PRS}$ = 36.57%, 95% CI = 35.89–37.26% versus $UKB_{PRS}$ = 33.23%, 95% CI = 32.56–33.91%) (Fig. 7d).

## Discussion
In this large multi-trait GWAS analysis, we show that cutaneous melanoma, 'any-cancer', pigmentation traits, autoimmune diseases and other serum metabolic biomarkers are genetically correlated

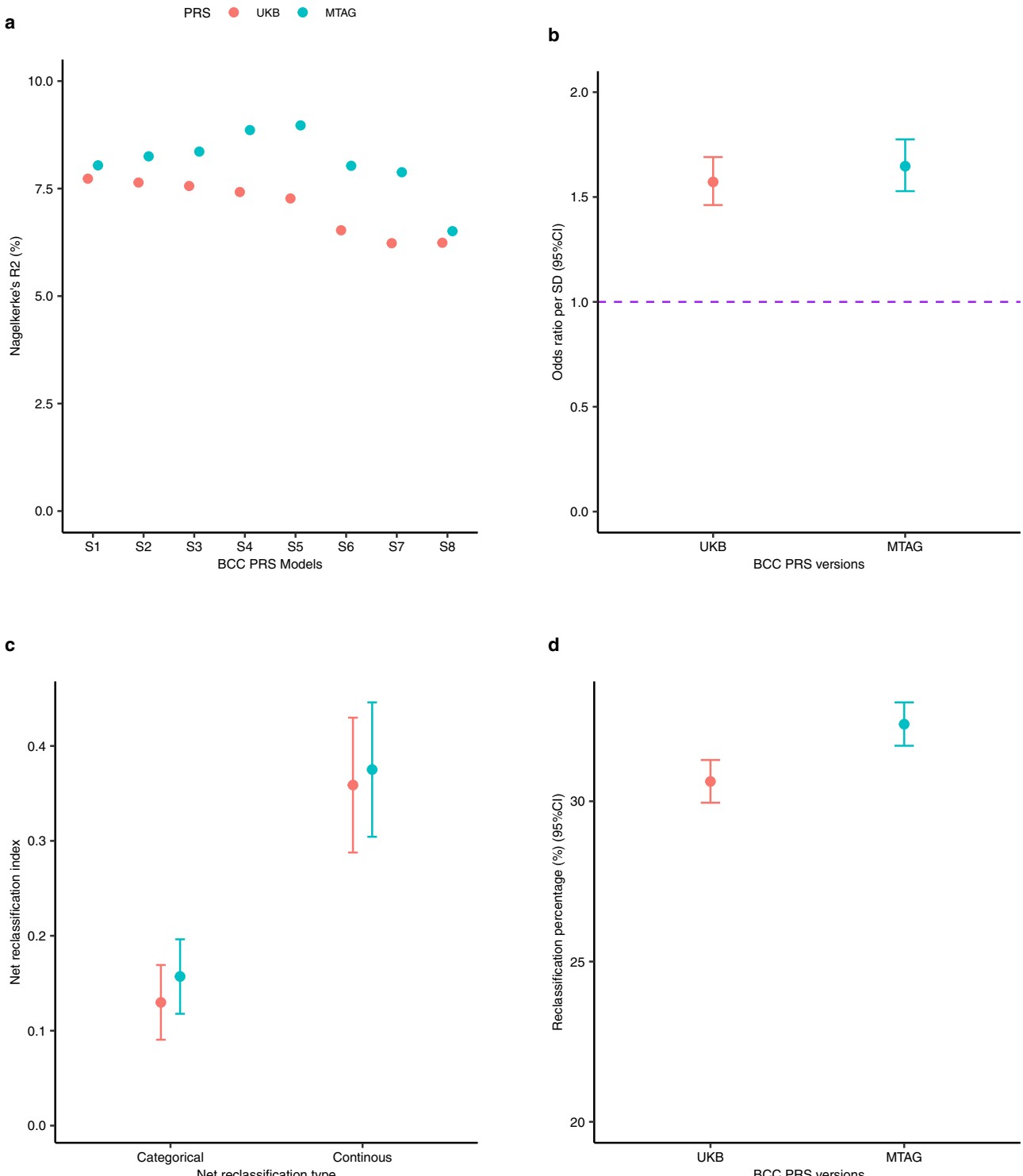

with BCC and SCC. We have leveraged this genetic correlation using the MTAG approach to identify 78 and 69 independent genome-wide significant loci for BCC and SCC risk, respectively, the most common skin cancers among fair-skinned people. Nineteen BCC and 15 SCC loci were previously unknown for any KC and replicated in the 23andMe cohort, indicating our study uncovers important findings relevant to KC biology.

First, we identify previously unknown loci in the pigmentation pathways for both BCC and SCC susceptibility. Due to the importance of sun exposure in keratinocyte cancer biology[29], several new loci for

BCC and SCC were linked to pigmentation traits, including skin colour, red hair, skin tanning response and sunburns. The gene-set analysis results also confirmed we identified biological pathways involved in melanin biosynthesis and DNA damage response.

Second, our study affirms the role of immune-regulatory processes and pathways in BCC and SCC susceptibility. We show that the previously unknown loci for BCC and SCC are implicated in immune-regulatory processes (Supplementary Information), including; HIV viral load modulation[30,31], innate immune response (through *IFIH1*)[32–34] and autoimmunity. These cellular immune responses are important in

**Fig. 7 | Validation and application of the basal cell carcinoma MTAG_PRS and UKB_PRS in participants in the Canadian Longitudinal Study on Aging (CLSA).** PRS refers to polygenic risk score, UKB- United Kingdom Biobank, MTAG multi-trait analysis of GWAS, CI confidence intervals, SD standard deviation, % percent and BCC basal cell carcinoma. The red colour represents the UKB PRS version whilst cyan indicates the MTAG-derived PRS. The error bars represent the 95% confidence interval in 6b (odds ratio, two-sided test), c (net reclassification improvement index) and d (percentage reclassified). **a** Validation of the BCC MTAG_PRS and UKB_PRS models to select the best performing index based on clumped SNPs at S1 ($P < 5 \times 10^{-8}$), S2 ($P < 10^{-7}$), S3 ($P < 10^{-6}$), S4 ($P < 10^{-5}$), S5 ($P < 10^{-4}$), S6 ($P < 10^{-3}$), S7 ($P < 10^{-2}$) and S8 ($P < 10^{-1}$) on the x-axis. The y-axis represents Nagelkerke's $R^2$ (%), a measure for model fitness. PRS model S1 and S5 are the optimal PRS models for UKB_PRS and MTAG_PRS, respectively, in a selected validation sample of CLSA ($N = 1911$ individuals). **b** Shows and compares the association between the UKB_PRS and MTAG_PRS and KC risk in CLSA ($N = 18,515$ individuals) expressed in odds ratios per standard deviation (y-axis) increase in the PRS, and adjusted for age, sex and the ancestral 10 PCs. **c** Illustrates that the MTAG_PRS performs better than the UKB_PRS based on both the categorical and continuous net reclassification improvement indices in CLSA ($N = 18,515$ individuals). **d** Compares the percentage of people reclassified to an appropriated KC risk group after adding the MTAG_PRS vs the UKB_PRS to a model with age, sex and 10 ancestral principal components in CLSA ($N = 18,515$); MTAG_PRS reclassified 36.57%, 95% CI = 35.89–37.26% of individuals compared to 33.23%, 95% CI = 32.56–33.91% by UKB_PRS. Source data are provided as a Source Data file.

cancer initiation and progression[35]. We also highlight a previously known locus (*CTLA4*) which is an immunotherapy target (anti-CTLA4 medication) in melanoma treatment[36]. Therefore, our identified loci implicated in immune response may be potential targets to improve immunotherapy for skin cancer. However, further functional genomic studies will be needed to establish their potential role in skin cancer prevention and treatment.

Third, immunosuppressive medication, including azathioprine and cyclosporin A have been implicated in BCC and SCC risk[37,38]. While we uncovered KC loci linked to immune-related medication use, including; anti-asthmatic inhalants and thyroid preparations[39], it is likely that medication-related loci underpinned here are just a proxy indicator for the autoimmune disease. Thus, these medications are unlikely to cause BCC or SCC. In addition, even if these diseases were all treated with drugs that greatly increased the risk of KC, they are (a) too rare to lead to a cryptic genetic correlation as large as what we see here e.g. for hypothyroidism (rg = −0.19, $P = 1.05 \times 10^{-4}$) (Supplementary Data 1) and (b) the genetic correlation e.g. for hypothyroidism was negative with BCC where a drug-induced cryptic overlap would give a positive genetic correlation.

Fourth, our study also highlights the potential role of cardiometabolic biomarkers in BCC/SCC risk. Besides the PUFA levels, whose causal association link with the BCC risk has been established through a Mendelian randomisation study[40], our results highlight a potential causal relationship between cardiometabolic biomarkers, including; diastolic and systolic blood pressure, lipids, serum glucose, cholesterol and adiposity, and the risk of BCC and SCC. As is the case for PUFA, downstream metabolism of these cardiometabolic biomarkers, such as lipids and cholesterol, results in oncogenic inflammatory biomarkers (e.g. prostaglandins E, thromboxane A2 and leukotriene B). However, some risk genetic variants or loci for the cardiometabolic pathway could be influencing BCC and SCC risk through already known pigmentation and immune-regulatory biological pathways e.g. rs1136165 in *CKB* and rs10774625 in *ATXN2*[41–44].

Fifth, we also unveil important genes with a potential role in BCC and SCC initiation and progression e.g. *FAP, CDKL1, MARK3, RAB11FIP2, GAB2, SUOX* and *SOX6*. Although some genetic variants within these genes have pleiotropic effects with pigmentation traits, the aforementioned genes have established roles in cancer cell proliferation, migration and invasion, and downregulation of apoptosis in melanoma, colorectal cancer and breast cancer[45–51]. Some of these loci are potential drug targets. For example, a previous study identified a potential drug 'PCC0208017' as an inhibitor of *MARK3*, suppressing glioma progression both in vitro and in vivo[52]. Fostamatinib, a drug used for treatment of chronic immune thrombocytopenia[53], is an inhibitor of *MARK3*[54]. Further studies are warranted to test these drugs for any anti-tumour activity in KC.

Our results further emphasise the shared biology between cutaneous melanoma and KC. In total, four previously unknown loci for BCC and SCC at *ATM, DSTYK, GPR98* and *SOX6* are known for CM[20,55,56]. Our MTAG results have also highlighted shared biology between BCC and SCC whereby almost half (7) of the previously unknown loci are

shared between BCC and SCC. However, our work also highlights loci distinct to either BCC (12) and SCC (8), indicating unique biological pathways (see results) for each cancer.

We also note the difference in the replication success between BCC and SCC. Given the relatively high genetic correlation between the two traits, similar replication results are expected. However, at a subset of loci, the input data may suggest that a particular SNP is only strongly associated with say, BCC but no SCC. Given we have substantially more input data on BCC than SCC, power may also play a part in the strength of the results, and replication success. We have previously shown that BCC is twice as heritable as SCC (SNP-heritability estimates for BCC = 13.1%, 95% CI = 9.7–16.5% versus 6.8%, 95% CI = 0.9–12.7% for SCC)[13], and it is more polygenic[8,57]. We believe the reasons contributed to the differences in replication success.

One strength of the MTAG method is the increase in statistical power to identify several loci that a standard single-trait GWAS would not have done. For example, using MTAG, we increased our sample size by 2.6 times and 8.3 times for BCC and SCC, respectively. Owing to the great improvement in statistical power, our MTAG-derived BCC PRS outperformed (for KC risk stratification) the one derived from a single-trait BCC GWAS. We and others have previously shown that the KC PRS generated from the general population effectively stratify them for KC risk and multiplicity[58–60]. The optimised MTAG-derived PRS is likely to improve KC risk stratification in high-risk subpopulations, as previously shown in solid organ transplant recipients.

One caveat with the MTAG approach is that it assumes that the genetic variants have a homogeneous effect across all the included traits so that the results are not driven by a certain trait to result in false positives[15]. Firstly, when we compared the genetic correlation (Supplementary Fig. 1), and the MTAG results (Supplementary Fig. 2) before and after excluding genomic regions (*HLA, ASIP, IRF4, MC1R, SLC45A2* and *CDKN2A*) with very large effect sizes for skin cancers and pigmentation traits, and there was a high concordance (Supplementary Fig. 2). Secondly, there was good replication of our results in an independent cohort, which counters concerns of false positives. In addition, in order to minimise biases arising from using several cohorts which might have phenotypes with different measures[15], we selected only traits where the magnitude of the genetic correlation was larger than 0.1 (or less than −0.1 for negatively correlated traits); we also required the correlation to at least reach nominal significance ($P < 0.05$), as a priori. Also, studies with small sample size were not considered, as including such traits would only negligibly increase our effective sample size.

In conclusion, leveraging the genetic correlation between skin cancers, autoimmune diseases, pigmentation traits and serum biochemistry biomarkers revealed previously unknown susceptibility loci for SCC and BCC, implicated in KC development and progression, pigmentation, cardiometabolic and immune-regulatory pathways. We also report an optimised PRS for effective risk stratification for KC, which could facilitate skin cancer surveillance in high-risk subpopulations such as transplantees.

## Methods

### Cohorts

**Discovery cohorts.** Participants that contributed to the phenotype-specific genome-wide association studies were of homogenous European ancestry drawn from different cohorts from Australia, Europe and America. While there was sample overlap across the included GWAS, MTAG adjusts and corrects for biases due to sample overlap[15]. The major cohorts used included; the UK Biobank (UKB)[18,19], QSkin Sun and Health Study (QSkin) (Olsen et al. 2012), eMERGE (dbGaP, study accession: phs000360.v3.p1) and GERA (dbGaP, study accession: phs000674.v3.p3), a melanoma meta-analysis consortium (Supplementary Information; Supplementary Table 2)[20] (dbGaP accession study code: phs001868.v1.p1), as well as publicly available GWAS summary statistics from international cohorts and consortium. Details for each cohort, including ethics oversight, are described in the Supplementary Information.

**Replication cohort: 23andMe Research Cohort.** 23andMe, Inc. is a direct-to-consumer genetic company that collected both self-reported phenotypes and genetic data from participants who provided informed consent and participated in the research online, under a protocol approved by the external Association for the Accreditation of Human Research Protection Programme (AAHRPP)- accredited Institutional Review Board (IRB), Ethical & Independent Review Services (E&I Review). The BCC cohort included 2,523,630 participants of European ancestry; 251,963 BCC cases and 2,271,667 controls, and 44.65% males. The SCC dataset included 2,529,399 participants of European ancestry; 134,700 SCC cases and 2,394,699 controls, and 44.65% males. Further details on data collection, validation, genotyping, imputation and quality control have been published before[8,57].

**BCC PRS application cohort: the Canadian Longitudinal Study on Aging (CLSA).** The Canadian Longitudinal Study on Aging (CLSA) is a prospective large population-based cohort in Canada comprising about 50,000 participants (45–85 years) randomly recruited between 2010 and 2015 from ten provinces[61,62]. More information about the cohort has been published elsewhere[61,62] and summarised here. It consists of two cohorts; the 'Tracking cohort' of ~20,000 participants recruited through a telephone questionnaire in ten provinces, and the "Comprehensive cohort" with ~30,000 individuals who provided data through an in-person questionnaire, clinical/physical tests and biological samples (e.g. for genetic data) in seven provinces.

In general, at baseline, information on relevant variants, including age and sex, were recorded, and participants were also asked whether they had been diagnosed with any cancer, including KC (yes/no), by a health professional. Between 2015 and 2018, the first follow-up assessment was conducted and participants were asked again if they had been diagnosed with cancer, and KC during the follow-up period. Thus, the CLSA dataset we used included the 'Baseline Comprehensive Dataset version 4.0' and 'Follow-up 1 Comprehensive Dataset version 1.0'. At the time of analysis, ~30,000 individuals had genetic data available, genotyped using 820 K UK Biobank Axiom Array (Affymetrix)[61], and imputed using the TopMed imputation server[63]. The CLSA is overseen by the Canadian Institutes of Health Research (CIHR) and its protocol has been reviewed and approved by 13 research ethics boards in Canada. All participants provided written informed consent.

Firstly, for purposes of validation and selection of the optimal PRS models (as described below in Stage 6 analysis) we randomly selected 1523 cancer-free controls and 388 prevalent KC cases at the baseline. Thus, our validation sample included 1911 participants with a mean age of 65.81 years (sd = 10.25) and 52.75% males.

Secondly, we tested the BCC PRSs in a second sample (unrelated to the validation dataset) of 18,933 participants of European ancestry, with a mean age of 61.80 years (sd = 9.84), followed up for a mean duration of 2.9 years (sd = 0.3) and 49.63% males. Only participants with complete data on age, sex, cancer status and KC diagnosis were

included. Thus, 18,139 controls with no history of any cancer (at follow up 1) and 794 participants who developed KC during follow-up.

### Statistical analysis

**Stage 1: GWAS for BCC, SCC and related traits.** We conducted two case-control GWAS using UKB data for BCC, $N = 307,684$ (20,791 cases and 286,893 controls) and SCC, $N = 294,294$ (7402 SCC cases and 286,892 controls) of European ancestry. We adjusted for age and sex as well as the first ten ancestral principal components (PCs) in order to control for biases from population stratification. We used Scalable and Accurate Implementation of GEneralised mixed model (SAIGE) software for the analysis since it controls for sample relatedness and case-control imbalance[25]. Analysis was restricted to single nucleotide polymorphism (SNPs) with minor allele frequency (MAF) >1% and an imputation quality score of 0.3. BCC/SCC cases were drawn from UK cancer registries. Further details on case ascertainment and definition are described in Supplementary Information.

In addition, we conducted GWAS for pigmentation traits (e.g. skin colour, hair colour, tanning response, skin burn, sunburn, etc.), all-cancer, autoimmune conditions, and blood biochemistry biomarkers (e.g. C-reactive protein, vitamin D, glucose, albumin, aspartate aminotransferase, gamma-glutamyl transferase, etc) using data from international cohorts including; UKB, QSkin, and GERA as described in Supplementary Information, Supplementary Data 2, 3. We also conducted GWAS on KC and all-cancer after accessing data from eMERGE (dbGaP, study accession: phs000360.v3.p1) and GERA (dbGaP, study accession: phs000674.v3.p3) cohorts respectively (Supplementary Information). We also accessed publicly available GWAS summary statistics e.g. for cutaneous melanoma[20], smoking[28], education attainment[27], body mass index[64], hypothyroidism, type 1 diabetes, rheumatoid arthritis[25] and vitiligo[26] (Supplementary Information, Supplementary Data 2, 3).

**Stage 2: Genetic correlation between BCC, SCC and related traits.** We used LDSC version 1.0.1[65], to compute the genetic correlation ($r_g$)[16] between BCC and a range of other traits, including; other skin cancer types, pigmentation traits, autoimmune traits and biochemistry biomarkers (recently released in the UKB). We then repeated this process for SCC instead of BCC. We used data from publicly available GWAS, as well as GWAS data from international cohorts of participants of European ancestry (conducted in stage 1 above). Traits with a statistically significant ($P < 0.05$) $r_g$ greater than 10% with either BCC or SCC were selected and included in the MTAG model (Fig. 1 and Supplementary Table 1). We further sought additional traits that were genetically correlated with BCC or SCC using data from the LD hub catalogue[17]. Out of about 700 phenotypes, no additional phenotypes were selected to be included in the final MTAG model.

In total, 22 traits, including the initial input BCC and SCC GWAS from different cohorts of European ancestry, met the inclusion criteria. The 22 genetically correlated traits included; BCC, SCC, skin colour, hair colour excluding red hair, hypothyroidism, type 1 diabetes, gamma-glutamyl transferase, aspartate aminotransferase, serum vitamin D levels, albumin, C-reactive protein and glucose in the UK Biobank[19], KC, red hair and mole count in the QSkin[21], KC in eMERGE (dbGaP, study accession: phs000360.v3.p1), all-cancer in GERA cohort (dbGaP, study accession: phs000674.v3.p3), melanoma risk as measured by the latest and largest melanoma risk gwas meta-analysis[20], vitiligo[26], education attainment[27] and smoking[28]. All the above studies excluded 23andMe, to enable us to utilise the 23andMe data as a replication set. Details on the phenotypic measurements and definitions are described in Supplementary Information and Supplementary Data 2, 3.

**Stage 3: Multi-trait analysis of GWAS summary statistics.** Next, using a total of 22 genetically correlated traits, we conducted a multi-phenotype analysis of GWAS summary statistics (generated at stage 1

analysis and selected in stage 2) using MTAG software version 1.0.8[15]. MTAG default settings were used. MTAG combines GWAS summary statistics for genetically correlated traits into a meta-analysis while accounting for genetic correlation, sample overlap, maximising power to identify loci associated with the trait(s) of interest (here BCC and SCC)[15]. MTAG generates trait-specific results for each phenotype included in the model. BCC and SCC GWAS summary data from UKB from stage 1 were included as trait 1 and 2, respectively in the model below;

MTAG model: BCC + SCC + melanoma + pigmentation traits
+ autoimmune traits + ....... + trait *n*.

After the quality control measures, the analysis was restricted to 5,301,239 SNPs common in all the 22 GWAS with a minor allele frequency of >1%, and no ambiguous alleles. MTAG boosts the statistical power of the single-trait GWAS[15]. We assessed the increase in the statistical power/effective sample size or the GWAS-equivalent sample size when MTAG was applied to the single-trait GWAS, by comparing the average chi-squared before and after MTAG for BCC and for SCC using the following formula recommended by the MTAG authors:[15]

$$(1 - \text{average } \chi^2 \text{ MTAG } output)/(1 - \text{average } \chi^2 \text{ MTAG } \_input)$$

Where MTAG input corresponds to the input for either BCC or SCC GWAS in the UKB dataset, and χ2 is chi-squared.

We took forward the a) BCC and b) SCC MTAG output summary statistic results for further post-GWAS analysis in stage 4 and replication in stage 5. BCC and SCC Manhattan plots are presented in Figs. 2 and 3, respectively.

**Stage 3.1: Sensitivity analyses.** MTAG assumes a homogeneous effect across all the included traits[15]. However, due to their strong association with some input traits, the following genomic regions were removed; *CDKN2A*, *SLC45A2*, *IRF4* and *HLA* for autoimmune, and *ASIP* and *MC1R* for pigmentation or CM, violate this assumption. We conducted sensitivity analyses excluding these regions before implementing our MTAG model. Using the stage 1 BCC GWAS summary statistics, we removed extended regions for *ASIP* on chromosome 20 (30–36 megabases (mb)), *MHC* regions on chromosome 6 (25–36 mb), and *MC1R* on chromosome 16 (87–90.3 mb). We also removed 2 mb around the most significant SNP in the following regions; rs12203592 (6:396321) in the *IRF4* region on chromosome 6, rs3731239 (9:21974218) in the *CDKN2A* region on chromosome 9, and rs16891982 (5:33951693) in the *SLC45A2* region on chromosome 5. We compared the genetic correlation between BCC/SCC before and after removing the genomic regions with known strong associations and high LD (Supplementary Fig. 2), before running the full MTAG model of 22 traits described above. The MTAG results with and without the above genomic regions were also compared (Supplementary Fig. 1).

**Stage 4: Post-GWAS analysis.** We used FUMA v.1.3.6[66], to identify independent, genome-wide significant SNPs and the genomic risk loci, and performed annotation of candidate SNPs in the genomic loci and functional gene mapping. We also conducted gene-based and pathway analyses using MAGMA v.1.7, as implemented in FUMA v.1.3.6[67]. For the gene pathway analysis, gene ontology (GO) and curated gene sets from MSigDB (v5.2)[68] were used and corrected for multiple testing. GWAS catalogue[69] and Open Targets platform[70] were used to annotate the loci and their relationship with other traits.

**Stage 5: Replication of the BCC and SCC MTAG results.** Next, we sought to replicate the BCC and SCC susceptibility loci in a large independent cohort using data from the 23andMe research cohort. For BCC, the replication cohort included 251,963 self-reported cases and 2,271,667 controls, while the SCC replication comprised 134,700 cases

and self-reported cases and 2,394,699 controls of European ancestry filtered to remove close relatives.

Previous studies have shown high accuracy of 23andMe BCC/SCC self-reported cases[8] and high genetic correlation ($r_g > 0.9$) between the histologically confirmed UKB BCC/SCC data and 23andMe data[13]. Age, sex, and population stratification using five PCs were adjusted for in both analyses in a logistic regression i.e.

$$\text{BCC or SCC} \sim \text{genotype} + \text{age} + \text{sex} + pc.0 + pc.1 + pc.2 + pc.3 + pc.4$$
$$+ v2\_\text{platform} + v3\_0\_\text{platform} + v3\_1\_\text{platform} + v4\_\text{platform}.$$

The V2 genotyping platform was a variant of the Illumina HumanHap550 + BeadChip with ~560,000 SNPs, including about 25,000 custom SNPs selected by 23andMe. The V3 platform included Illumina OmniExpress + BeadChip with ~950,000 SNPs and custom content SNPs. The V4 is the current and fully custom array of ~950,000 SNPs and includes a lower redundancy subset of V2 and V3 SNPs[71].

The BCC results were adjusted for a genomic control inflation factor λ = 1.286. The equivalent inflation factor for 1000 cases and 1000 controls λ1000 = 1.001, and for 10000, λ10000 = 1.006. In a similar way, the SCC results were adjusted for a genomic control inflation factor λ = 1.172. The equivalent inflation factor for 1000 cases and 1000 controls λ1000 = 1.001, and for 10000, λ10000 = 1.007. Thus, this inflation factor was not concerning as it is proportional to the large sample size[72]. We also explored any evidence of inflation in the discovery GWAS by assessing the LDSC intercept[73], which showed no inflation (not substantially above 1) for both BCC (LDSC intercept = 0.96, 95% CI = 0.94–0.99) and SCC (LDSC intercept = 0.77, 95%CI = 0.75–0.79).

We also compared the concordance of the effect sizes (log OR) for the MTAG results versus the replication results (Fig. 4b, d). We further analysed the number of loci that replicated at a genome-wide significant level ($P = 5.0 \times 10^{-8}$), after multiple testing correction (i.e. Bonferroni correction $P = 6.49 \times 10^{-4}$ for BCC; correcting for 77 loci, and $P = 7.24 \times 10^{-4}$ for SCC; correcting for 69 loci) and at a nominal $P = 0.05$.

**Stage 6: Development and validation of the BCC Polygenic Risk Score in a selected sample of participants in CLSA.** To construct two comparable polygenic risk scores (PRSs) for BCC, we separately used the BCC MTAG output (generated in stage 3) and the UKB BCC single-trait GWAS (generated in stage 1) summary statistics as the discovery data sets. MTAG[15] drops SNPs with extremely significant associations with any input trait, which resulted in a number of previously reported pigmentation-associated SNPs being dropped from the model. Hence in both the MTAG and UKB discovery GWAS summary statistics, we also included two functional SNPs (rs1805007 for *MC1R*, and rs12203592 for *IRF4*) that would otherwise have been dropped in the PRS using the weights from a previously published BCC PRS[74]. They are removed during the MTAG analysis as it filters out SNPs strongly ($P < 10 \times 2.22^{-308}$) associated with input traits, but this same strong association confirms they are important for a PRS for BCC. A sensitivity analysis results excluding these SNPs, and still, the MTAG BCC PRS reclassified skin cancer cases to a higher risk group (41.27%) better than the single BCC PRS (37.95%).

Next, using autosomal, non-ambiguous, and bi-allelic SNPs overlapping in the CLSA cohort (MTAG discovery = 5,300,872 SNPs and UKB discovery = 5,300,868 SNPs), we performed LD clumping based on ($r^2 = 0.005$ and LD window = 5000 kb, $P = 1$) to yield 62,494 and 62,884 independent SNPs for MTAG$_{\text{PRS}}$ and UKB$_{\text{PRS}}$ models respectively. PLINK 1.90b6[75] for clumping. Using the clumped independent SNPs above, we generated PRS models at varying *p* value thresholds i.e. S1 ($P < 5 \times 10^{-8}$), S2 ($P < 10^{-7}$), S3 ($P < 10^{-6}$), S4 ($P < 10^{-5}$), S5 ($P < 10^{-4}$), S6 ($P < 10^{-3}$), S7 ($P < 10^{-2}$) and S8 ($P < 10^{-1}$) in validation sample of 1911 participants split from the CLSA cohort using log odds ratio (from the respective discovery GWAS; MTAG or UKB) as weights. PLINK2 (v2.00a3LM 5 May 2021 release)[75] was used for generating the PRS scores.

For both MTAG and UKB PRS models, we used Nagelkerke's $R^{2}$[76], a metric for model fitness used for selecting the optimal model. We computed the $R^2$ by comparing the model fitness between models with PRSs (BCC-MTAG$_{PRS}$ *or* UKB$_{PRS}$ + age + sex + 10 Pcs) and a null model using predictABEL package[77] in R software version 4.0.2[78].

**Stage 7: Applying BCC polygenic risk score and keratinocyte cancer risk prediction in the Canadian longitudinal study of aging.** To determine the ability of our MTAG GWAS data to predict skin cancer, we used 18,933 participants of European ancestry with data on KC risk in the Canadian Longitudinal Study of Aging (CLSA). We included 18,139 controls with no history of any cancer (both at baseline and follow-up) and 794 cases who developed KC during the 2.9 years (on average) follow-up following baseline recruitment. Separate BCC and SCC data were unavailable in this cohort, and as ~80% of KC cases are BCC cases[79], we tested the performance MTAG$_{PRS}$ vs UKB$_{PRS}$ derived for BCC to predict the risk of KC.

Using PLINK2 (v2.00a3LM 5 May 2021 release)[75], we generated individual scores for CLSA participants for both the BCC MTAG$_{PRS}$ and UKB$_{PRS}$ weighted by their respective effect sizes (log odds ratios). The genetic scores were standardised to a variance of 1 in order to interpret the associations as odds ratio per standard deviation increase in the PRS. We compared the performance of the two BCC PRSs (MTAG$_{PRS}$ vs UKB$_{PRS}$) based on the magnitude of the association (odds ratios) and the net reclassification improvement for KC risk using R version 4.0.2[78]. For net reclassification improvement, we compared the net reclassification index and the percentage of the participants who got reclassified to an appropriate risk group/tertile i.e. the low risk (bottom tertile), moderate risk (middle tertile), and high risk (top tertile) after adding the MTAG$_{PRS}$ vs UKB$_{PRS}$ to the base model containing age, sex and the ten PCs.

### Reporting summary

Further information on research design is available in the Nature Portfolio Reporting Summary linked to this article.

## Data availability

The full GWAS summary statistics generated in this study have been deposited in the NHGRI-EBI GWAS Catalogue under accession code GCST90137411 for BCC and GCST90137412 for SCC. The PRS generated in this paper are provided with this paper in Supplementary Data 6, 7. The BCC and SCC independent genome-wide significant SNPs (both for the discovery of MTAG results and replication from 23andMe) generated in this study are provided with this paper in Supplementary Data 4, 5 files, respectively. Source data for the figures are provided with this paper in the Source Data File. Genotype and phenotype data for the UK Biobank are available through application via https://www.ukbiobank.ac.uk/, for the Canadian Longitudinal Study on Aging (CLSA) at www.clsa-elcv.ca, and for QSkin through application to Prof. David Whiteman (David.Whiteman@qimrberghofer.edu.au), the principal investigator. Data from the Resource for Genetic Epidemiology Research on Aging (GERA) Cohort (dbGap accession phs000674.v3.p3), and the Electronic Medical Records and Genomics Network (eMERGE) (dbGaP, study accession: phs000360.v3.p1) can be accessed from dpGAP. The cutaneous melanoma GWAS summary statistics used in this paper were from ref. 20, are publicly available from dbGap (accession study code: phs001868.v1.p1). The following trait-specific GWAS summary statistics were also used in this paper and are publicly available through the following consortia or resources; educational attainment by ref. 27 (downloadable from the SSGAC website http://ssgac.org/Data.php), smoking (cigarettes per day) by ref. 28 (downloadable from the GSCAN Consortium website https://genome.psych.umn.edu/index.php/GSCAN), and autoimmune traits from ref. 25 (available at ftp://share.sph.umich.edu/UKBB_SAIGE_HRC/). Source data are provided with this paper.

## Code availability

The custom code used to generate the key results in this study can be freely accessed at https://github.com/mathiasS-hub/KC_MTAG_NatComm_Code.

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

## Acknowledgements
This study was supported by a programme grant (APP1073898) and a project grant (APP1063061) from the Australian National Health and Medical Research Council (NHMRC). S.M. and D.C.W. are supported by Research Fellowships from the NHMRC. M.S. was supported by the Australian Government Research Training Programme (RTP) and the Faculty of Health Scholarship at Queensland University of Technology, Australia. This study was conducted using data from UK Biobank (application number 25331), QSkin Sun and Health Study (Australia), Canadian Longitudinal Study on Aging (CLSA), 23andMe (USA), the Electronic Medical Records and Genomics Network (eMERGE) (USA), the Resource for Genetic Epidemiology Research in Adult Health and Aging (GERA) (USA) and other publicly available data from other consortia. We thank the research participants and the employees of 23andMe for partly making this work possible. The eMERGE data used in this study included samples from Vanderbilt University, the University of Washington, Marshfield Clinic, Mayo Clinic, and Northwestern University. We acknowledge each contributing cohort separately in the Supplementary Information (Supplementary Note on Acknowledgments). The GERA data were accessed from dbGaP (dbGaP accession phs000674.v3.p3). We acknowledge the submitter(s) of this study. Data came from a grant, the Resource for Genetic Epidemiology Research in Adult Health and Aging (RC2 AG033067; Schaefer and Risch, PIs), awarded to the Kaiser Permanente Research Programme on Genes, Environment, and Health (RPGEH) and the UCSF Institute for Human Genetics. The RPGEH was supported by grants from the Robert Wood Johnson Foundation, the Wayne and Gladys Valley Foundation, the Ellison Medical Foundation, Kaiser Permanente Northern California and the Kaiser Permanente National and Northern California Community Benefit Programmes.

The opinions expressed in this manuscript are the author's own and do not reflect the views of the Canadian Longitudinal Study on Aging or any affiliated institution. This research was made possible using the data/bio-specimens collected by the Canadian Longitudinal Study on Aging (CLSA). Funding for the Canadian Longitudinal Study on Aging (CLSA) is provided by the Government of Canada through the Canadian Institutes of Health Research (CIHR) under grant reference: LSA 94473 and the Canada Foundation for Innovation. This research has been conducted using the CLSA dataset [Baseline Comprehensive Dataset version 4.0, Follow-up 1 Comprehensive Dataset version 1.0], under Application Number 190225. The CLSA is led by Drs. Parminder Raina, Christina Wolfson and Susan Kirkland.

## Author contributions
Conceptualisation: M.S., M.H.L. and S.M.; Data curation: M.S., M.H.L., D.C.W., C.M.O. and S.M.; Formal analysis: M.S.; Funding acquisition: S.M., M.H.L. and D.C.W.; Investigation: M.S., M.H.L., D.C.W., C.M.O. and S.M.; Methodology: M.S., M.H.L., and S.M.; Project administration: M.S., M.H.L. and S.M.; Resources: M.H.L., D.C.W. and S.M.; Software: M.S.; Supervision: S.M. and M.H.L.; Visualisation: M.S.; Writing—original draft preparation: M.S.; Writing—revision of subsequent drafts: M.S., S.M., M.H.L., J.-S.O., P.G., D.C.W., C.M.O. and P.F. Replication of the results: P.F., 23andme Research Team. All authors contributed to the final version of the manuscript.

## Competing interests
Pierre Fontanillas, Chao Tian and the 23andMe Research Team are employed by and hold stock or stock options in 23andMe, Inc. The remaining authors declare no competing interests.

## Additional information

## The 23andMe Research Team

Stella Aslibekyan[4], Adam Auton[4], Elizabeth Babalola[4], Robert K. Bell[4], Jessica Bielenberg[4], Katarzyna Bryc[4], Emily Bullis[4], Daniella Coker[4], Gabriel Cuellar Partida[4], Devika Dhamija[4], Sayantan Das[4], Sarah L. Elson[4], Teresa Filshtein[4], Kipper Fletez-Brant[4], Pierre Fontanillas ®[4], Will Freyman[4], Pooja M. Gandhi[4], Karl Heilbron[4], Barry Hicks[4], David A. Hinds[4], Ethan M. Jewett[4], Yunxuan Jiang[4], Katelyn Kukar[4], Keng-Han Lin[4], Maya Lowe[4], Jey McCreight[4], Matthew H. McIntyre[4], Steven J. Micheletti[4], Meghan E. Moreno[4], Joanna L. Mountain[4], Priyanka Nandakumar[4], Elizabeth S. Noblin[4], Jared O'Connell[4], Aaron A. Petrakovitz[4], G. David Poznik[4], Morgan Schumacher[4], Anjali J. Shastri[4], Janie F. Shelton[4], Jingchunzi Shi[4], Suyash Shringarpure[4], Vinh Tran[4], Joyce Y. Tung[4], Xin Wang[4], Wei Wang[4], Catherine H. Weldon[4], Peter Wilton[4], Alejandro Hernandez[4], Corinna Wong[4] & Christophe Toukam Tchakouté[4]

