## [Peer review file · Nature Communications]

REVIEWER COMMENTS

Reviewer #1 (Remarks to the Author): Expert in BCC and SCC genetics, genomics, and risk factors

In this manuscript, Seviiri and colleagues identify and replicate novel genetic loci for the keratinocyte carcinomas basal and squamous cell carcinoma. This work contributes to the ongoing efforts to identify genomic opportunities for patient risk stratification, which may enable appropriate use of healthcare resources for skin cancer screening.

A strength of this study is the use of large and robust datasets including the UK Biobank for derivation and 23andMe for validation. Previous studies in this area have been limited by either small size or lack of granular patient-derived information such as skin/hair/eye color, or sun exposure/protection behavior. As a result, previous studies in the field have predominantly reported on the association of genes of pigmentation such as MC1R and OCA2 with skin cancer risk. The use of larger datasets as well as incorporation of multitrait analysis of GWAS (MTAG) enables the authors to address the overlap between susceptibility loci for keratinocyte carcinoma as well as the closely related pigmentation traits and others.

- Editors please note that I am not experienced with the execution of MTAG and cannot comment on the method specifics beyond the innovative and appropriate use of this tool, and that the previously reported GWAS included in the metaanalysis (lines 85-104) are robust and representative of the field.

Several of the novel loci identified in this study were predictably associated with pigmentation including hair color and skin propensity to tan vs. burn. As

More interesting are the novel reported loci in immune response. Both BCC and SCC are known to be highly immunogenic and both respond to immunotherapy with PD1 inhibition; it would be helpful if the authors could expand on discussion of the potential for these loci to further improve immunotherapy for either prevention or treatment of these cancers.

In the discussion of the novel immune loci, the authors relate them to previously published pathways for antiviral immune response and autoimmunity. Rather than suggesting shared biology between KC and "immune-related viruses" (line 433- and aren't all viruses immune-related?) it may be more accurate to describe the role of these loci in cellular immunity, which targets both cancerous and viral-infected cells.

The authors similarly describe loci associated with certain medications or autoimmunity. It would be more interesting to look at the actual immune system biology and suggest where there may be

anticancer immune processes that overlap with autoimmunity. Please comment on the known risk of immunosuppression for KC and whether an association with autoimmunity/upregulated immune response is paradoxical or expected due to imbalance in immune responses.

Do any of the potential loci for cancer initiation and progression suggest molecular targets for therapy? Currently the only targeted molecular inhibitor for KC are the hedgehog pathway inhibitors for BCC; previous efforts to use EGFR inhibition for cSCC have been unpredictable. An oral or topical targeted inhibitor would be a valuable asset for KC, particularly in high risk transplant recipients who are not candidates for checkpoint inhibition.

What is the hypothesis proposed for the resulting loci in cardiometabolic pathways?

The authors additionally report a polygenic risk score for BCC that predicts incidence of basal cell carcinoma, and validate this in the Canadian Longitudinal Study on Aging. It is not quite clear why no PRS score is reported for SCC, only for BCC- or why the BCC PRS is applied to the overall KC risk- please clarify.

Quite a few polygenic risk scores have been published for KC, yet none have been put into broad clinical use. Can the authors please comment on this score in context- is the main utility to adjust future GWAS and continue to identify rare variant, or is there a role for PRS in skin cancer screening? If so, please comment on the cost-benefit of PRS and additional information that would be provided over clinical risk stratification- particularly as the genes with largest effect sizes can be easily predicted by simply asking a patients' hair/eye color. How does this PRS tool compare in PPV to the very simple Qskin risk calculator?

<https://publications.qimrberghofer.edu.au/p/qimr/qskinriskcalculator>

Minor comment:

In the methods section, please comment on whether BCC/SCC in the UK and 23andMe cohorts are derived from patient self-report or are drawn cancer registry or physician/billing records.

If space permits, it would be appropriate to include some comment on the generalization of these results to patients of non-European ancestry. Clinically, identifying the rare patient of color who will go on to develop skin cancer and may benefit from screening is an unmet medical need. The authors may

wish to review Jorgenson E et al Genetic ancestry, skin pigmentation, and the risk of cutaneous squamous cell carcinoma in Hispanic/Latino and non-Hispanic white populations. *Commun Biol.* 2020;3(1):765.

Reviewer #2 (Remarks to the Author): Expert in cancer genetics, multi-trait analysis, and GWAS

The work presented here by the authors represents a nice and huge effort to refine the Skin Cancer Genetics for susceptibility and their classification using a genetic correlation wide-analysis.

The authors expand previous work on SCC and BBC genetic to a large catalogue of at risk related conditions considered to have a role in BCC/ SCC pathogenesis, and identifies new loci of susceptibility 19 (BCC) and 15 (SCC) giving insights in their role in Skin Cancer. Moreover, they define a new PRS for BCC offering showing a higher power classification than that derived from a Single GWAs.

The strategy is not novel but offers an improvement in the BCC patient classification using germline genomic variation. The methodology is sound and offers a possibility to expand the field of pleiotropic genetic analysis a powerful approach to analyse the complex inheritance of common conditions using large genetic datasets published everywhere.

I have some major points that should be addressed. Some of these concerns are related to the presentation of the study, even if are not scientific relevant, it should be ameliorated in the final form to better understand the validity and biases of the such approach.

1. The large number of datasets used in the analysis make difficult valuate the biases that could arise from their use in this study. The cohort/study used in each moment (or/and its exclusion) should be carefully identified in the main manuscript, since is not easy to follow. The explanation of the cohorts used, are sometimes presented with different names, or only with the bibliographic reference, having a puzzle impression. An example is in the ST9, in which names / numbers of the studies are not easy to identify related to main manuscript or other tables. Even if most of the information could be find in the supplementary material, it is fragmentary and difficult to follow. I think that an important part of this paper should give light on the MATG utility. The understanding of the nature of the different sources included their biases and the manner how the authors identified and managed it is important for this paper. A special effort to present this should be done.

2. The genetic analysis presented here (all analysis from correlation to GWAs) are based in data from different studies, and the former selection of variants is based on MAF and the info score of imputation. However, in any part of the MS there is any reference to the imputation method used by the different cohorts, neither the array-used. I will like to have, at least, (1) a description of the imputation methods and used arrays in the summary description of the studies included, and (2) some attention to these important points in the discussion (e.g about the bias effect of this heterogeneity) to give some clue of how to control this in future studies.

3. The description of the new findings that the authors are summarized systematically by area (from pag. 14 to 19), but I think that it should be presented in the same way for all the areas (as is presented in the cardiometabolic pathway) since it makes difficult follow the findings and their relevance regarding the BCC / SCC. And this is one of the presented outputs of the paper, beyond increasing the number of loci.

4. What is the different impact of MTAG results on BCC or SCC? since differences in severity between both types of KC, I will like to know, how the MTAG could help to differentiate better both entities? Or how this helps to understand both entities. For instance, you observed differences in the replication success for BCC and SCC, do you have any idea of these differences? Is something related to phenotypes included? Is an artefact due to diagnosis/ number or other technical issues? A little discussion is merited in the discussion section.

5. Regarding PRS, (1) the PRS method is not presented in the methods section (Pag20), it starts with the validation. Clearly a better explanation is needed to be understood the approach for all readers. (2) Comparing both PRS MTAG / SINGLE, only a few SNPs are shared (64) between both PRSs, what is the explanation for this? The "type of genes" involved are different? Is this related to power issues? Even if presented in the supplementary files, this should be stated in the main MS, and discussed. (3) I will appreciate information of the genes included in the PRS, even if the annotation is for the nearest gene, to get some insight in the functional impact of this new PRS, Did you perform any analysis regarding functional categories includes and the differences if any?. (4) why there is not presented a PRS for SCC? It will be interesting to see its behaviour? These differences could give additional insights in the differences between both SK entities. (5) what is the reason to re-include those SNPs that were not considered in the MTAG analysis (in the text say they drop the analysis) for the MTAG-PRS? Are these SNPs essentials for the better behaviour of the new presented PRS? A sensitivity analysis without these SNPs should be performed, if not done already.

6. In the Section 5: Is not clear to me what was the adjusted method for a genomic control inflation. What does mean the equivalent inflation factor? A better explanation is needed, as well some discussion about the high genomic control inflation factor λ observed (near 1,2)

Minor points

1. In the MS, several Bonferroni correction are performed. Sometimes is hard to know for what we are correcting. Please, stated for what it accounts in the different points that it is used. Sometimes is not clear. Please review in the text.
2. To better appreciate the new findings, you present in the paper, it will be nice to show in different colour the new regions arising from the MTAG analysis, in the Manhattan Plots (Figure 2 and 3). This will offer a visual context of they.
3. EAF definition is lacking in the Table 1 and Table 2.
4. The proposed calculation of the effective sample size needs to be explained better. What is the rationale of the proposed method?
5. In the Stage 5: Replication of the BCC and SCC MTAG results, is not clear to me what is the Platform?
BCC or SCC \sim genotype + age + sex + pc.0 + pc.1 + pc.2 + pc.3 + pc.4 + v2_platform + v3_0_platform + v3_1_platform + v4_platform.

Reviewer #4 (Remarks to the Author): Expert in BCC and SCC genetics

The manuscript describes a complex MTAG approach to investigate susceptibility loci for basal cell carcinoma (BCC) and squamous cell carcinoma (SCC). The authors identified 19 novel SNPs for BCC and 15 for SCC and develop a PRS for BCC. The topic is of considerable interest to the readers, and only minor weaknesses are present.

In detail:

1. The abstract lacks a conclusion that might help understand the relevance of the study.
2. In the Results section, it might be helpful to summarize together the most relevant data regarding SNPs. Much of the Result section indeed includes speculation to compare the results with data from the literature (i.e., page 14 line 12 to page 19 line 8.). The authors should avoid lengthy discussion and include it in the Supplementary Results. In addition, they could include the most important information reported in the text for the selected SNPs in a new table.
3. I would suggest standardizing the way the authors refer to nevus counts. In Figure 1, "Nevus" is listed, whereas in Supplementary Table 1, "Mole counts" is listed.
4. Tables 1 and 2. Please include the meaning of the abbreviation EAF.

5. Page 16, line 1. Authors should standardize the way they refer to genes (i.e. if they use the abbreviation for genes, it should also be used for interleukin-2 receptor).
5. Additional table 4 lacks the abbreviation for acronyms.
6. In Supplementary Table 6, it is not clear which pathways refer to BCC or SCC. Please construct the table to better associate pathways with tracts (BCC or SCC or both).
5. Authors should indicate in all figure legends (both standard and supplemental) what the abbreviations mean.

RESPONSE TO REVIEWERS' COMMENTS

REVIEWER COMMENTS

Reviewer #1 (Remarks to the Author): Expert in BCC and SCC genetics, genomics, and risk factors

In this manuscript, Seviiri and colleagues identify and replicate novel genetic loci for the keratinocyte carcinomas basal and squamous cell carcinoma. This work contributes to the ongoing efforts to identify genomic opportunities for patient risk stratification, which may enable appropriate use of healthcare resources for skin cancer screening.

A strength of this study is the use of large and robust datasets including the UK Biobank for derivation and 23andMe for validation. Previous studies in this area have been limited by either small size or lack of granular patient-derived information such as skin/hair/eye color, or sun exposure/protection behavior. As a result, previous studies in the field have predominantly reported on the association of genes of pigmentation such as MC1R and OCA2 with skin cancer risk. The use of larger datasets as well as incorporation of multitrait analysis of GWAS (MTAG) enables the authors to address the overlap between susceptibility loci for keratinocyte carcinoma as well as the closely related pigmentation traits and others.

- Editors please note that I am not experienced with the execution of MTAG and cannot comment on the method specifics beyond the innovative and appropriate use of this tool, and that the previously reported GWAS included in the metaanalysis (lines 85-104) are robust and representative of the field.

Several of the novel loci identified in this study were predictably associated with pigmentation including hair color and skin propensity to tan vs. burn. As

More interesting are the novel reported loci in immune response. Both BCC and SCC are known to be highly immunogenic and both respond to immunotherapy with PD1 inhibition; it would be helpful if the authors could expand on discussion of the potential for these loci to further improve immunotherapy for either prevention or treatment of these cancers.

Response

We agree with the reviewer that some of the loci reported in immune response may be potential drug targets to improve immunotherapy. For example our results identified a previously known locus (*CTLA4*) which is a immunotherapy target for anti-CTLA4 medication in melanoma treatment. Although the reported new novel loci have a role in immune response, further functional genomics studies will be needed to establish their potential role in skin cancer prevention and treatment. This is beyond the scope of this study.

In the discussion of the novel immune loci, the authors relate them to previously published pathways for antiviral immune response and autoimmunity. Rather than suggesting shared biology

between KC and "immune-related viruses" (line 433- and aren't all viruses immune-related?) it may be more accurate to describe the role of these loci in cellular immunity, which targets both cancerous and viral-infected cells.

Response

We thank the reviewers for their advice. We have addressed this in the discussion and in the "Description of the novel loci" in the **Supplementary Information**, and focused on cellular immunity.

The authors similarly describe loci associated with certain medications or autoimmunity. It would be more interesting to look at the actual immune system biology and suggest where there may be anticancer immune processes that overlap with autoimmunity. Please comment on the known risk of immunosuppression for KC and whether an association with autoimmunity/upregulated immune response is paradoxical or expected due to imbalance in immune responses.

Response

Immunosuppression is a well known risk factor for KC. For example, organ transplant recipients with chronic immunosuppression have up to 100-fold and 20-fold increased risk for squamous cell carcinoma (SCC) and basal cell carcinoma (BCC), respectively compared to the general population (Lindelöf et al. 2000). But we agree that the SCC and autoimmunity genetic correlation (rg) reported/used is paradoxical relative to the expected direction of effect. To some extent the genetic correlation is negative for BCC - suggesting propensity to autoimmunity is weakly associated with a lower genetic risk for BCC, but for SCC the point estimate rgs are positive (Supplementary Table 1). This is probably that the rg estimates for SCC are imprecise - they are non sig in SCC. However, this still means if e.g. type 1 diabetes were included in the SCC MTAG they would have been combined with SCC with a positive genetic correlation (rg).

Do any of the potential loci for cancer initiation and progression suggest molecular targets for therapy? Currently the only targeted molecular inhibitor for KC are the hedgehog pathway inhibitors for BCC; previous efforts to use EGFR inhibition for cSCC have been unpredictable. An oral or topical targeted inhibitor would be a valuable asset for KC, particularly in high risk transplant recipients who are not candidates for checkpoint inhibition.

Response

Yes, e.g. a recent study identified a potential drug "PCC0208017" as an inhibitor of *MARK3*, suppressing glioma progression both *in vitro* and *in vivo* (Li et al. 2020). Further studies are warranted to test this drug for any anti-tumour activity in KC. Fostamatinib, a drug used for treatment of chronic immune thrombocytopenia (Connell and Berliner 2019), is an inhibitor of *MARK3* (Rolf et al. 2015). The *eQTL* direction supports the idea that inhibition of *MARK3* is a good idea. rs1136165 T allele

(for MARK3) is associated with a positive beta or effect estimate, so increased risk (Table 1). The same T allele is associated with reduced *MARK3* expression in GTEX skin - sun exposed. This supports the idea that inhibiting *MARK3* is the right approach. However, comprehensive genomic functional studies will be needed to investigate further these loci and the possible drugs targeting them to treat KC. In the discussion, we have included a paragraph on some of the possible drug targets.

What is the hypothesis proposed for the resulting loci in cardiometabolic pathways?

Response

One of the possible hypothesis is that some of these loci e.g. *FADS1/2* regulate downstream metabolism cardiometabolic factors e.g. cholesterol, polyunsaturated fatty acids (PUFAs) etc which results oncogenic inflammatory biomarkers (prostaglandins E, thromboxane A2, and leukotriene B) (Azrad, Turgeon, and Demark-Wahnefried 2013). For example, using a Mendelian randomisation approach, we have previously shown that PUFAs, whose metabolism is regulated by *FADS1/2* is causally associated with BCC risk (Seviiri, Law, Ong, Gharakhani, Nyholt, Olsen, et al. 2021b)

The authors additionally report a polygenic risk score for BCC that predicts incidence of basal cell carcinoma, and validate this in the Canadian Longitudinal Study on Aging. It is not quite clear why no PRS score is reported for SCC, only for BCC- or why the BCC PRS is applied to the overall KC risk- please clarify.

Response

We thank the reviewers for bringing this concern to our attention. The Canadian Longitudinal Study on Aging, which we used to test the PRS only recorded “KC” as phenotype; no specific information BCC and SCC. Given that ~80% of “overall KC” cases in the general population are BCC, we felt it was more appropriate to use only the BCC PRS in the absence of specific information on SCC and BCC. We would have liked to test an SCC PRS on SCC risk in an independent dataset, but the power for a SCC PRS would have been greatly reduced when applied to the combined KC phenotype. In the **Methods Section (Statistical Analysis; Stage 7: BCC Polygenic Risk Score and Keratinocyte Cancer Risk Prediction in the Canadian Longitudinal Study of Aging)** we have updated this sentence “*Separate BCC and SCC data were unavailable in this cohort, and as ~80% of KC cases are BCC cases (Ciążyńska et al. 2021) we tested the performance $MTAG_{PRS}$ vs UKB_{PRS} derived for BCC to predict the risk of KC.*”

Quite a few polygenic risk scores have been published for KC, yet none have been put into broad clinical use. Can the authors please comment on this score in context- is the main utility to adjust future GWAS and continue to identify rare variant, or is there a role for PRS in skin cancer screening? If so, please comment on the cost-benefit of PRS and additional information that would be provided over clinical risk stratification- particularly as the genes with largest effect sizes can be easily predicted by simply asking a patients' hair/eye color. How does this PRS tool compare in PPV to the very simple Qskin risk calculator?

<https://publications.qimrberghofer.edu.au/p/qimr/qskinriskcalculator>

Response

In our previous work we have shown that KC PRSs generated from the general population work well in organ transplant recipients (OTRs) to stratify them for KC risk and multiplicity (Seviiri, Law, Ong, Gharakhani, Nyholt, Hopkins, et al. 2021; Seviiri, Law, Ong, Gharakhani, Nyholt, Olsen, et al. 2021a). OTRs are at ultra-high risk of developing KC (up to 100-fold) (Lindelöf et al. 2000). We showed that the KC PRSs improve prediction of KC risk over and above the traditional clinical skin cancer risk factors by up to 3% (e.g. BCC, AUC = 0.73 vs. 0.70), and 19% (for BCC) and 18% (for SCC) of the SOTRs are reclassified in a high/medium/low risk scenario when the PRS is added to clinical stratification models.(Seviiri, Law, Ong, Gharakhani, Nyholt, Hopkins, et al. 2021) For clinical utility purposes, the PRS would be used in conjunction with the clinical risk factors to accurately triage OTRs for KC risk and design appropriate skin cancer screening strategies and other preventive strategies for each risk group.

We agree that the role for a PRS in clinical management for BCC and SCC is currently unclear and an area of active research. As noted above, we have evidence in the OTR setting that adding a PRS can improve on traditional risk, but a more detailed cost-benefit analysis of the clinical utility of a PRS versus traditional risk scores, and modelling/trialling how a PRS might be implemented in screening or risk stratification (population wide, or in restricted settings such as for OTR) is outside of the scope of this paper. We are working with a collaborator, David Whiteman, to more fully compare KC, BCC or SCC PRS with the QSkin risk calculator.

Minor comment:

In the methods section, please comment on whether BCC/SCC in the UK and 23andMe cohorts are derived from patient self-report or are drawn from cancer registry or physician/billing records.

Response:

We thank the reviewers for bringing this to our attention. We have updated this information in the Methods Section for the UKB; under *Statistical analysis, Stage 1: GWAS for BCC, SCC and related traits*. For 23andme this information was included under *Statistical analysis, Stage 5: Replication of the BCC and SCC MTAG results*.

If space permits, it would be appropriate to include some comment on the generalization of these results to patients of non-European ancestry. Clinically, identifying the rare patient of color who will go on to develop skin cancer and may benefit from screening is an unmet medical need. The authors may wish to review Jorgenson E et al Genetic ancestry, skin pigmentation, and the risk of cutaneous squamous cell carcinoma in Hispanic/Latino and non-Hispanic white populations. *Commun Biol.* 2020;3(1):765.

Response

We agree that addressing the transferability of PRS developed in European ancestry GWAS to non-European population is an important and constantly evolving field. Adapting KC/BCC/SC PRS to Hispanic/Latino and non-Hispanic white populations requires access to suitable populations with genetic and phenotypic risk data that are not currently available to us. We are in discussions with suitable biobanks (e.g. the Global Biobank Meta-analysis Initiative) but these project applications are still in early days.

Reviewer #2 (Remarks to the Author): Expert in cancer genetics, multi-trait analysis, and GWAS

The work presented here by the authors represents a nice and huge effort to refine the Skin Cancer Genetics for susceptibility and their classification using a genetic correlation wide-analysis.

The authors expand previous work on SCC and BBC genetic to a large catalogue of at risk related conditions considered to have a role in BCC/ SCC pathogenesis, and identifies new loci of susceptibility 19 (BCC) and 15 (SCC) giving insights in their role in Skin Cancer. Moreover, they define a new PRS for BCC offering showing a higher power classification than that derived from a Single GWAs.

The strategy is not novel but offers an improvement in the BCC patient classification using germline genomic variation. The methodology is sound and offers a possibility to expand the field of pleiotropic genetic analysis a powerful approach to analyse the complex inheritance of common conditions using large genetic datasets published everywhere.

I have some major points that should be addressed. Some of these concerns are related to the presentation of the study, even if are not scientific relevant, it should be ameliorated in the final form to better understand the validity and biases of the such approach.

1. The large number of datasets used in the analysis make difficult valuate the biases that could arise from their use in this study. The cohort/study used in each moment (or/and its exclusion) should be carefully identified in the main manuscript, since is not easy to follow. The explanation of the cohorts used, are sometimes presented with different names, or only with the bibliographic reference, having a puzzle impression. An example is in the ST9, in which names / numbers of the studies are not easy to identify related to main manuscript or other tables. Even if most of the information could be find in the supplementary material, it is fragmentary and difficult to follow. I think that an important part of this paper should give light on the MATG utility. The understanding of the nature of the different sources included their biases and the manner how the authors identified and managed it is important for this paper. A special effort to present this should be done.

Response

All the cohorts and respective traits are explained in the **Supplementary Information** (Methods) and in **Supplementary Table 3**. We have included a title and more descriptive information for Supplementary Table 9 (ST9).

In order to minimise biases arising from using several cohorts which might have phenotypes with different measures (Turley et al. 2018), we selected only traits where the magnitude of the genetic correlation was larger than 0.1 (or less than -0.1 for negatively correlated traits); we also required the correlation to at least reach nominal significance ($P < 0.05$), as a prior. In addition, studies with small sample size were not considered, as including such traits would only negligibly increase our effective sample size.

In order to be sure that our results were not false positives, we sought replication of the results from a large and independent cohort; 23andMe, Inc (BCC: 251,963 cases and 2,271,667 controls, and SCC: 134,700 cases and 2,394,699 controls). The MTAG authors recommend replication of the MTAG findings to ensure findings are robust (Turley et al. 2018). In the event, all of the 19 novel loci for BCC and 15 for SCC were replicated in our independent cohort.

Lastly, in order to rule out that the BCC and SCC effect sizes were not unduly influenced by the other 20 traits, we compared the **MTAG versus UK GWAS effect estimates**. We found a high concordance with a Pearson's correlation of 0.93 (95% confidence interval [CI]= 0.89-0.96) for BCC and a Pearson's correlation of 0.71, 95% CI=0.57-0.81 for SCC.

We have elaborated the above points in the discussion.

2. The genetic analysis presented here (all analysis from correlation to GWAs) are based in data from different studies, and the former selection of variants is based on MAF and the info score of imputation. However, in any part of the MS there is any reference to the imputation method used by the different cohorts, neither the array-used. I will like to have, at least, (1) a description of the imputation methods and used arrays in the summary description of the studies included, and (2) some attention to these important points in the discussion (e.g about the bias effect of this heterogeneity) to give some clue of how to control this in future studies.

Response

We thank the reviewer 3 for raising these concerns. 1) We have updated **Supplementary Table 3** to include summary information on the genotyping arrays and reference arrays used. Detailed information on the same was provided in the **Supplementary Information**, in the **Methods section** for each cohort. We have also provided further information on imputation of the eMERGE Cohort. In summary, the following arrays were used for genotyping; UK BiLEVE Axiom and UKB Axiom

Arrays for UKB, Illumina GSA for QSkin, Illumina Human660W-Quad_v1_A array for eMERGE, Affymetrix Axiom arrays for GERA. The majority of the cohorts used the HRC reference panel for imputation (**Supplementary Table 3**). In addition to the HRC, the UK10K reference panel was also used for UKB. Therefore, it is highly likely that the set of imputed variants were consistent.

2) In order to control for any form of heterogeneity arising from the different definitions of the phenotypes, the MTAG authors recommend using traits that are genetically correlated (Turley et al. 2018). As explained before, we selected only traits where the magnitude of the genetic correlation was larger than 0.1 (or less than -0.1 for negatively correlated traits) with BCC or SCC; we also required the correlation to at least reach nominal significance ($P < 0.05$), as a prior. We have added an explanation **in the second last paragraph in the Discussion**.

3. The description of the new findings that the authors are summarized systematically by area (from pag. 14 to 19), but I think that it should be presented in the same way for all the areas (as is presented in the cardiometabolic pathway) since it makes difficult follow the findings and their relevance regarding the BCC / SCC. And this is one of the presented outputs of the paper, beyond increasing the number of loci.

Response

We thank the reviewers for this suggestion. We have updated the immune response, KC development and initiation, and pigmentation pathways to highlight which loci were distinct to BCC, SCC or shared. This section has also been moved to Supplementary Information as suggested by reviewer 4.

4. What is the different impact of MTAG results on BCC or SCC? since differences in severity between both types of KC, I will like to know, how the MTAG could help to differentiate better both entities? Or how this helps to understand both entities. For instance, you observed differences in the replication success for BCC and SCC, do you have any idea of these differences? Is something related to phenotypes included? Is an artefact due to diagnosis/ number or other technical issues? A little discussion is merited in the discussion section.

Response

The main essence of using MTAG is to leverage the genetic overlap between traits to improve statistical power to identify novel loci for the traits in contention (Turley et al. 2018). For traits like SCC and BCC where the genome-wide genetic correlation is relatively high, most loci will be shared and similar results are expected from MTAG. However, at a subset of loci the input data may suggest that a particular SNP is only strongly associated with say BCC but no SCC. Given we have substantially more data on BCC than SCC, power may also play a part in the strength of the results (and hence the replication) for each type of KC. Moreover, we have previously shown that BCC is twice as heritable as SCC (SNP-heritability estimates for BCC = 13.1%, 95% CI = 9.7–16.5% versus

6.8%, 95% CI = 0.9–12.7% for SCC) (Liyanage et al. 2019), and it is more polygenic (Chahal, Wu, et al. 2016; Chahal, Lin, et al. 2016). We believe those reasons contributed to the differences in replication success.

However, our MTAG results have enhanced our understanding of shared biology between BCC and SCC whereby almost half (7) of the noci loci are shared between BCC and SCC. However, MTAG has also highlighted loci distinct to either BCC (12) and SCC (8), indicating unique biological pathways (See results) for each cancer. In the discussion, we have added these similarities and differences.

5. Regarding PRS, (1) the PRS method is not presented in the methods section (Pag20), it starts with the validation. Clearly a better explanation is needed to be understood the approach for all readers. (2) Comparing both PRS MTAG / SINGLE, only a few SNPs are shared (64) between both PRSs, what is the explanation for this? The “type of genes” involved are different? Is this related to power issues? Even if presented in the supplementary files, this should be stated in the main MS, and discussed. (3) I will appreciate information of the genes included in the PRS, even if the annotation is for the nearest gene, to get some insight in the functional impact of this new PRS, Did you perform any analysis regarding functional categories includes and the differences if any?. (4) why there is not presented a PRS for SCC? It will be interesting to see its behaviour? These differences could give additional insights in the differences between both SK entities. (5) what is the reason to re-include those SNPs that were not considered in the MTAG analysis (in the text say they drop the analysis) for the MTAG-PRS? Are these SNPs essentials for the better behaviour of the new presented PRS? A sensitivity analysis without these SNPs should be performed, if not done already.

Response

5 (1) The validation part on page 24 refers to the results from validation of the PRS. The PRS method including development and validation, and testing are presented in the Methods; Stage 6 and 7 respectively (pages 39-41).

5(2) - There are two reasons; i) The MTAG results have more power than the single BCC analysis and therefore, it has more SNPs reaching significance. ii) Different lead SNPs chosen in a given genomic region so while specific SNPs might be 'missing' a different one in linkage disequilibrium is present. Therefore, based on the nearest gene analysis, 154 SNPs (**Supplementary Table 10**) overlap between the MTAG and the SINGLE BCC PRS; that is 56.41% of the SINGLE PRS SNPs. The correlation of the effect size for the PRS SNPs across the two sets is consistent or high (e.g. for the overlapping 154 SNPs Pearson's correlation = 0.94, 95% CI=0.92-0.96, $P < 2.2 \times 10^{-16}$), meaning the extra MTAG SNPs are consistent but just better powered. We have added an explanation in the discussion.

5(3) - We have included in the Supplementary Tables 7 and 8 information on the possible nearest gene.

5(4) - Regarding not presenting the SCC PRS, please refer to our response to Question 3 under reviewer 1.

5(5) - The two SNPs; rs12203592 and rs1805007 are re-introduced because they are well known functional SNPs affecting skin cancer susceptibility. Rs12203592 (*IRF4*) acts through the immune regulatory pathway whilst rs1805007 (*MC1R*) influences skin cancer risk through pigmentation pathways. They are removed during the MTAG analysis as it filters out SNPs strongly ($P < 10 \times 10^{-8}$) associated with input traits, but this same strong association confirms they are important for a PRS for BCC/SCC. We have included a sensitivity analysis results excluding these SNPs, and still the MTAG BCC PRS reclassified skin cancer cases to a higher risk group (41.27%) better than the SINGLE BCC PRS (37.95%).

6. In the Section 5: Is not clear to me what was the adjusted method for a genomic control inflation. What does mean the equivalent inflation factor? A better explanation is needed, as well some discussion about the high genomic control inflation factor λ observed (near 1,2)

Response

The best method of assessing inflation is the linkage disequilibrium score regression (LDSC) intercept (normally between 0 and 1) as this only inflates substantially above 1 if there is e.g. double counted samples or population stratification (Bulik-Sullivan et al. 2015). The genomic control lambda method used by 23andme is a reasonable method of assessing inflation in small sample set but as we get to large sample sizes (such as in the data in this study), lambda gets inflated, even in a well conducted gwas (where there is e.g. no double counting of samples and no population stratification concerns) (Yang et al. 2011). So 23andMe reported the lambda scaled as if you only have 1000 cases and 1000 controls etc, showing that the inflation factor is not concerning as it is proportional to their large sample size. Thus, 23andMe corrected for the lambda and they also reported the equivalent inflation factor.

To confirm that there was no inflation in the discovery GWAS, we have included the LDSC intercept results for BCC (LDSC intercept = 0.96, 95%CI = 0.94 -0.99) and SCC (LDSC intercept = 0.77, 95%CI = 0.75 -0.79) which were not substantially above 1. We have added an explanation under **Methods: Stage 5: Replication of the BCC and SCC MTAG results.**

Minor points

1. In the MS, several Bonferroni correction are performed. Sometimes is hard to know for what we are correcting. Please, stated for what it accounts in the different points that it is used. Sometimes is not clear. Please review in the text.

Response

Thank you for highlighting this. In the Methods section (Stage 5: Replication of the BCC and SCC MTAG results) we have included the number of tests/loci that were corrected for in each case of BCC and SCC.

2. To better appreciate the new findings, you present in the paper, it will be nice to show in different colour the new regions arising from the MTAG analysis, in the Manhattan Plots (Figure 2 and 3). This will offer a visual context of they.

3. EAF definition is lacking in the Table 1 and Table 2.

Thank you for bringing this to our attention, we have added the definition for EAF (effect allele frequency) in **Tables 1 and 2**.

4. The proposed calculation of the effective sample size needs to be explained better. What is the rationale of the proposed method?

Thank you for bringing this to our attention, we have provided a better explanation in the **Methods Section (Stage 3: Multi-trait analysis of GWAS summary statistics**

) “We assessed the increase in the statistical power/effective sample size or the GWAS-equivalent sample size when MTAG was applied to the single trait GWAS, by comparing the average chi-squared before and after MTAG for BCC and for SCC using the following formula recommended by the MTAG authors (Turley et al. 2018):

$$(1 - \text{average } \chi^2 \text{ MTAG output}) / (1 - \text{average } \chi^2 \text{ MTAG input})$$

Where MTAG input corresponds to the input for either BCC or SCC GWAS in the UKB dataset, and χ^2 is chi -squared statistic.

5. In the Stage 5: Replication of the BCC and SCC MTAG results, is not clear to me what is the Platform? BCC or SCC ~ genotype + age + sex + pc.0 + pc.1 + pc.2 + pc.3 + pc.4 + v2_platform + v3_0_platform + v3_1_platform + v4_platform.

Platform refers to the genotyping platform used by 23andMe. We have replaced this with a better word “genotype batch effects” since that is what was controlled for. Details on the four platforms are published elsewhere (Tian et al. 2017). It is stated that “ The V1 and V2 genotyping platforms were variants of the Illumina HumanHap550 + BeadChip with ~ 560,000 SNPs, including about 25,000 custom SNPs selected by 23andMe. The V3 platform included Illumina OmniExpress + BeadChip with ~ 950,000 SNPs and custom content SNPs. The V4 is the current and fully custom array of ~ 950,000 SNPs and includes a lower redundancy subset of V2 and V3 SNPs” (Tian et al. 2017)

Reviewer #4 (Remarks to the Author): Expert in BCC and SCC genetics

The manuscript describes a complex MTAG approach to investigate susceptibility loci for basal cell carcinoma (BCC) and squamous cell carcinoma (SCC). The authors identified 19 novel SNPs for BCC and 15 for SCC and develop a PRS for BCC. The topic is of considerable interest to the readers, and only minor weaknesses are present.

In detail:

1. The abstract lacks a conclusion that might help understand the relevance of the study.

Response

We have included a conclusion in the abstract. "In conclusion, leveraging the genetic correlation between skin cancers, autoimmune diseases and pigmentation traits revealed novel susceptibility loci for SCC and BCC, as well as produced a powerful and optimised PRS for KC risk stratification. Novel loci are implicated in keratinocyte cancer development and progression, pigmentation, cardiometabolic pathways, and immune-regulatory pathways".

2. In the Results section, it might be helpful to summarize together the most relevant data regarding SNPs. Much of the Result section indeed includes speculation to compare the results with data from the literature (i.e., page 14 line 12 to page 19 line 8.). The authors should avoid lengthy discussion and include it in the Supplementary Results. In addition, they could include the most important information reported in the text for the selected SNPs in a new table.

Response

Thank you for this suggestion, we have included **Figures 5a and 5b** summarising the biological pathways for the loci, and we have moved the detailed description of the loci and the biological pathways to the **Supplementary Information**.

3. I would suggest standardizing the way the authors refer to nevus counts. In Figure 1, "Nevus" is listed, whereas in Supplementary Table 1, "Mole counts" is listed.

Response

Thank you for bringing this up. We have now used naevus count throughout the paper.

4. Tables 1 and 2. Please include the meaning of the abbreviation EAF.

Response

Thank you for bringing this to our attention, we have added the definition for EAF (effect allele frequency) in **Tables 1 and 2**.

5. Page 16, line 1. Authors should standardize the way they refer to genes (i.e. if they use the abbreviation for genes, it should also be used for interleukin-2 receptor).

Response

We have named all the genes by their abbreviations including *IL2RA* for the interleukin-2 receptor alpha chain.

6. Additional table 4 lacks the abbreviation for acronyms.

Response

Thank you for identifying this. We have provided acronyms for all the supplementary tables.

7. In Supplementary Table 6, it is not clear which pathways refer to BCC or SCC. Please construct the table to better associate pathways with tracts (BCC or SCC or both).

Response

Thank you for identifying this error. **Supplementary Table 6** has now been updated with clear information on biological pathways associated with each cancer.

8. Authors should indicate in all figure legends (both standard and supplemental) what the abbreviations mean.

Response

Thank you for identifying this. We have defined all the abbreviations in all the figures.

REFERENCES

- Azrad, Maria, Chelsea Turgeon, and Wendy Demark-Wahnefried. 2013. "Current Evidence Linking Polyunsaturated Fatty Acids with Cancer Risk and Progression." *Frontiers in Oncology* 3 (September): 224.
- Bulik-Sullivan, Brendan K., Po-Ru Loh, Hilary K. Finucane, Stephan Ripke, Jian Yang, Schizophrenia Working Group of the Psychiatric Genomics Consortium, Nick Patterson, Mark J. Daly, Alkes L. Price, and Benjamin M. Neale. 2015. "LD Score Regression Distinguishes Confounding from Polygenicity in Genome-Wide Association Studies." *Nature Genetics* 47 (3): 291–95.
- Chahal, Harvind S., Yuan Lin, Katherine J. Ransohoff, David A. Hinds, Wenting Wu, Hong-Ji Dai, Abrar A. Qureshi, et al. 2016. "Genome-Wide Association Study Identifies Novel Susceptibility Loci for Cutaneous Squamous Cell Carcinoma." *Nature Communications* 7 (July): 12048.
- Chahal, Harvind S., Wenting Wu, Katherine J. Ransohoff, Lingyao Yang, Haley Hedlin, Manisha Desai, Yuan Lin, et al. 2016. "Genome-Wide Association Study Identifies 14 Novel Risk Alleles Associated with Basal Cell Carcinoma." *Nature Communications* 7 (August): 12510.
- Ciążyńska, Magdalena, Grażyna Kamińska-Winciorek, Dariusz Lange, Bogumił Lewandowski, Adam Reich, Martyna Sławińska, Marta Pabianek, et al. 2021. "The Incidence and Clinical Analysis of Non-Melanoma Skin Cancer." *Scientific Reports* 11 (1): 4337.
- Connell, Nathan T., and Nancy Berliner. 2019. "Fostamatinib for the Treatment of Chronic Immune Thrombocytopenia." *Blood*. <https://doi.org/10.1182/blood-2018-11-852491>.
- Li, Fangfang, Zongliang Liu, Heyuan Sun, Chunmei Li, Wenyan Wang, Liang Ye, Chunhong Yan, Jingwei Tian, and Hongbo Wang. 2020. "PCC0208017, a Novel Small-Molecule Inhibitor of MARK3/MARK4, Suppresses Glioma Progression in Vitro and in Vivo." *Acta Pharmaceutica Sinica B*. <https://doi.org/10.1016/j.apsb.2019.09.004>.
- Lindelöf, B., B. Sigurgeirsson, H. Gäbel, and R. S. Stern. 2000. "Incidence of Skin Cancer in 5356 Patients Following Organ Transplantation." *The British Journal of Dermatology* 143 (3): 513–19.
- Liyanage, Upekha E., Matthew H. Law, Xikun Han, Jiyuan An, Jue-Sheng Ong, Puya Gharahkhani, Scott Gordon, et al. 2019. "Combined Analysis of Keratinocyte Cancers Identifies Novel Genome-Wide Loci." *Human Molecular Genetics* 28 (18): 3148–60.
- Rolf, Michael G., Jon O. Curwen, Margaret Veldman-Jones, Cath Eberlein, Jianyan Wang, Alex Harmer, Caroline J. Hellawell, and Martin Braddock. 2015. "In Vitro Pharmacological Profiling of R406 Identifies Molecular Targets Underlying the Clinical Effects of Fostamatinib." *Pharmacology Research & Perspectives* 3 (5): e00175.
- Seviiri, Mathias, Matthew H. Law, Jue Sheng Ong, Puya Gharahkhani, Dale R. Nyholt, Peter Hopkins, Daniel Chambers, et al. 2021. "Polygenic Risk Scores Stratify Keratinocyte Cancer Risk among Solid Organ Transplant Recipients with Chronic Immunosuppression in a High Ultraviolet Radiation Environment." *The Journal of Investigative Dermatology* 141 (12): 2866–75.e2.
- Seviiri, Mathias, Matthew H. Law, Jue Sheng Ong, Puya Gharahkhani, Dale R. Nyholt, Catherine M. Olsen, David C. Whiteman, and Stuart MacGregor. 2021a. "Polygenic Risk Scores Allow Risk Stratification for Keratinocyte Cancer in Organ-Transplant Recipients." *The Journal of Investigative Dermatology* 141 (2): 325–33.e6.
- . 2021b. "Polyunsaturated Fatty Acid Levels and the Risk of Keratinocyte Cancer: A Mendelian Randomization Analysis." *Cancer Epidemiology, Biomarkers & Prevention: A Publication of the American Association for Cancer Research, Cosponsored by the American Society of Preventive Oncology* 30 (8): 1591–98.
- Tian, Chao, Bethann S. Hromatka, Amy K. Kiefer, Nicholas Eriksson, Suzanne M. Noble, Joyce Y. Tung, and David A. Hinds. 2017. "Genome-Wide Association and HLA Region Fine-Mapping Studies Identify Susceptibility Loci for Multiple Common Infections." *Nature*

Communications 8 (1): 599.

Turley, Patrick, Raymond K. Walters, Omeed Maghzian, Aysu Okbay, James J. Lee, Mark Alan Fontana, Tuan Anh Nguyen-Viet, et al. 2018. "Multi-Trait Analysis of Genome-Wide Association Summary Statistics Using MTAG." *Nature Genetics* 50 (2): 229–37.

Yang, Jian, the GIANT Consortium, Michael N. Weedon, Shaun Purcell, Guillaume Lettre, Karol Estrada, Cristen J. Willer, et al. 2011. "Genomic Inflation Factors under Polygenic Inheritance." *European Journal of Human Genetics*. <https://doi.org/10.1038/ejhg.2011.39>.

REVIEWERS' COMMENTS

Reviewer #1 (Remarks to the Author):

I appreciate the authors' thoughtful responses to review. I have no further comments.

Reviewer #2 (Remarks to the Author):

The authors have addressed most of the points required.

As last comment, I only want to remark, that; (1) the PRS method should be presented before the validation part to have a more meaningful reading, (2) differences in replication of the methods for BCC and SCC, possibly due to power-differences as suggest the authors, should be stated clearly in the text.

I recommend the publication of the paper.

Reviewer #4 (Remarks to the Author):

The manuscript has been improved. The Results section is focused on the most relevant data, and the specific description of SNPs has been well summarized in the Supplementary Information. I really appreciated the two new figures, Figure 5a and 5b.

RESPONSE TO REVIEWERS

REVIEWERS' COMMENTS

Reviewer #1 (Remarks to the Author):

I appreciate the authors' thoughtful responses to review. I have no further comments.

Reviewer #2 (Remarks to the Author):

The authors have addressed most of the points required.

As last comment, I only want to remark, that; (1) the PRS method should be presented before the validation part to have a more meaningful reading, (2) differences in replication of the methods for BCC and SCC, possibly due to power-differences as suggest the authors, should be stated clearly in the text.

I recommend the publication of the paper.

Part 1)- We thank the reviewer for looking at this further. However, the PRS method is already presented before the validation and testing part (Stage 5 and 7 statistical analysis). The earlier parts where the PRS is mentioned only **describe the Cohort** where the PRS was validated and tested, but not the actual method.

Part 2) - We have included this paragraph in the discussion. “We also note the difference in the replication success between BCC and SCC. Given the relatively high genetic correlation between the two traits, similar replication results are expected. However, at a subset of loci the input data may suggest that a particular SNP is only strongly associated with say BCC but no SCC. Given we have substantially more input data on BCC than SCC, power may also play a part in the strength of the results, and replication success. We have previously shown that BCC is twice as heritable as SCC (SNP-heritability estimates for BCC = 13.1%, 95% CI = 9.7–16.5% versus 6.8%, 95% CI = 0.9–12.7% for SCC) (Liyanage et al. 2019), and it is more polygenic (Chahal, Wu, et al. 2016; Chahal, Lin, et al. 2016). We believe reasons contributed to the differences in replication success.”

Reviewer #4 (Remarks to the Author):

The manuscript has been improved. The Results section is focused on the most relevant data, and the specific description of SNPs has been well summarized in the Supplementary Information. I really appreciated the two new figures, Figure 5a and 5b.